# Morphology-dependent entry kinetics and spread of influenza A virus

Sarah Peterl [ID] [1,2], Carmen M Lahr [ID] [1,2,9], Carl N Schneider[1,2], Janis Meyer[3], Xenia Podlipensky[1,2], Vera Lechner [ID] [1,2], Maria Villiou[4,5,10], Larissa Eis[6], Steffen Klein[1,2,11], Charlotta Funaya [ID] [6], Elisabetta Ada Cavalcanti-Adam[7,12], Frederik Graw[8], Christine Selhuber-Unkel [ID] [4,5], Karl Rohr[3] & Petr Chlanda [ID] [1,2 ✉]

## Abstract

**Influenza A viruses (IAV) display a broad variety of morphologies ranging from spherical to long filamentous virus particles. These diverse phenotypes are believed to allow the virus to overcome various immunological and pulmonary barriers during entry into the airway epithelium, and to influence the viral entry pathway. Notably, laboratory-adapted IAV strains predominantly adopt a spherical form, yet the factors driving this preference as well as the factors favoring filamentous morphology in physiological settings remain unclear. To address this, we generated fluorescent reporter viruses with identical surface glycoproteins but distinct morphologies and developed a correlative light and scanning electron microscopy workflow. This enabled us to investigate the impact of viral morphology on spread, and to identify conditions favoring either form. Our findings demonstrate that filamentous IAV spread significantly slower in various cell lines, consistent with delayed entry kinetics and in-cell cryo-electron tomography, explaining the predominance of spherical forms in laboratory-adapted strains. Cellular junction integrity, neuraminidase activity, and mucin do not inhibit IAV spread in a morphology-dependent manner. However, filamentous virions confer a selective advantage under neutralizing-antibody pressure against hemagglutinin.**

**Keywords** Filamentous influenza A virus; Mucin; Neutralizing antibodies; In situ cryo-ET; CLEM
**Subject Category** Microbiology, Virology & Host Pathogen Interaction

See also: EA Bruce

## Introduction

The influenza A virus (IAV) is an important human respiratory virus with a broad host range and primary reservoir in wild aquatic birds. In recent years, avian IAV has shown an alarming increase in spread across the globe and several spillovers to other species including domestic and wild-life mammals were detected resulting in high mortality rates (Baechlein et al, 2023; Cruz et al, 2023; Elsmo et al, 2023; Honce and Schultz-Cherry, 2023). To establish an infection in the human respiratory tract, IAV spreads through contact and aerosol and must overcome several pulmonary barriers, such as mucus and surfactant layer (Le Sage et al, 2023; LeMessurier et al, 2020). Morphologically, IAV is highly variable, ranging from spherical viruses of 100 nm in diameter to filamentous particles of more than 20 μm in length (Dadonaite et al, 2016). Filamentous IAV is consistently found in human and animal isolates. This was also shown by electron microscopy (EM) of viral isolates from the last H1N1 pandemic in 2009 (Nakajima et al, 2010; Neumann et al, 2009) and from H5N1 avian viruses (Arai et al, 2019). In contrast, spherical particles are more prevalent in lab-adapted strains. Serial passaging of clinical isolates in embryonated chicken eggs or Madin-Darby canine kidney (MDCK) cells frequently leads to a loss of filamentous morphology. Conversely, serial infection with spherical IAV in Guinea pigs has been shown to result in the emergence of filamentous virus particles (Seladi-Schulman et al, 2013).

IAV morphology is genetically dictated by the M segment (Roberts et al, 1998), which is required for virus-like particle assembly (Chlanda et al, 2015). This segment encodes for the M1 protein, which forms a helical layer underneath the viral envelope (Peukes et al, 2020), and for the M2 ion channel implicated in membrane scission (Rossman et al, 2010). Previous studies showed that the formation of filamentous virions depends on the actin cytoskeleton and Rab11-positive sorting endosomes (Bruce et al, 2010; Roberts et al, 1998; Simpson-Holley et al, 2002). Although it

[1]Schaller Research Groups, Department of Infectious Diseases, Virology, Medical Faculty, Heidelberg University, Heidelberg, Germany. [2]BioQuant Centre for Quantitative Biology, Heidelberg University, Heidelberg, Germany. [3]Biomedical Computer Vision Group, BioQuant, IPMB, Heidelberg University, Heidelberg, Germany. [4]Institute for Molecular Systems Engineering and Advanced Materials (IMSEAM), Heidelberg University, Heidelberg, Germany. [5]Max Planck School Matter to Life, Heidelberg University, Heidelberg, Germany. [6]Electron Microscopy Core Facility, Heidelberg University, Heidelberg, Germany. [7]Department of Cellular Biophysics, Max Planck Institute for Medical Research, Heidelberg, Germany. [8]Department of Internal Medicine 5, Hematology and Oncology, Friedrich-Alexander-Universität Erlangen-Nürnberg and Universitätsklinikum Erlangen, Erlangen, Germany. [9]Present address: Luxembourg Centre for Systems Biomedicine (LCSB), University of Luxembourg, Esch-sur-Alzette, Luxembourg. [10]Present address: Max Planck Institute for Polymer Research, Mainz, Germany. [11]Present address: Molecular Systems Biology Unit, European Molecular Biology Laboratory (EMBL), Heidelberg, Germany. [12]Present address: Cellular Biomechanics, University of Bayreuth, Bayreuth, Germany. ✉E-mail: petr.chlanda@bioquant.uni-heidelberg.de

has been shown that a single mutation in the M segment can lead to a morphological switch from spherical to filamentous and vice versa (Elleman and Barclay, 2004), IAV morphology is not solely encoded in the genome. The high morphological variety within a single IAV clone was proposed to be driven by the tuneable assembly process in response to environmental pressure (Vahey and Fletcher, 2019b). In addition, even in the scarcity of M1 and M2 proteins, the morphology of virions is maintained (Bourmakina and Garcia-Sastre, 2005). Regardless of morphology, infectious IAV incorporates only one set of a genome physically separated into eight viral ribonucleoprotein complexes (vRNPs), positioned in the leading end of budding virions (Calder et al, 2010; Chou et al, 2012). The assembly of long filamentous viruses presumably requires more time and larger amounts of hemagglutinin (HA) and matrix protein 1 (M1). As previously shown, NP:M1/HA ratios in filamentous particles are significantly lower than in spherical particles (Roberts et al, 1998). Laboratory-adapted spherical viruses are assumed to minimize the number of their structural proteins to encapsulate eight vRNPs during adaptation to cell culture.

The filamentous morphology has the advantage of providing a larger surface containing significantly more HA proteins required for entry. It is known that the IAV morphology dictates the route of entry. Filamentous virions predominantly enter by macropinocytosis, while spherical virions enter via clathrin-mediated endocytosis (CME) (de Vries et al, 2011; Matlin et al, 1981; Rossman et al, 2012). From the standpoint of viral fitness, high morphological variability within a virus population may therefore positively contribute to virus entry by exploiting more entry routes. Previous EM studies showed that small filamentous virions remain intact during cell entry. However, in vitro data revealed that filamentous virions disintegrate into spherical particles at endosomal pH (Rossman et al, 2012), indicating that large filaments undergo more complex uncoating.

Previous studies demonstrated that filamentous morphology is important for IAV spread and transmissibility. Replacement of the M segment of the spherical, non-transmissive A/Puerto Rico/8/34 (H1N1) virus with that of the pandemic, filamentous IAV isolate A/Netherlands/602/2009 (H1N1) yields a virus with filamentous morphology which has indistinguishable transmissibility to wild-type A/Netherlands/602/2009, as shown in a guinea pig transmission model (Campbell et al, 2014).

Overall, the importance of IAV morphological heterogeneity is often overlooked in both in vitro and in vivo IAV studies. The current model states that the filamentous morphology increases virus fitness at higher cell entry pressure (Vahey and Fletcher, 2019b). However, it is not fully understood why viruses have adapted to reduce morphological variability and minimize their shape to spheres in cell culture. In addition, the exact components of the pulmonary barriers that can be overcome by high morphological variability and filamentous morphology in physiological settings remain unidentified.

In our study, we generated reporter viruses encoding polymerase acidic proteins (PA) tagged with mScarlet that carry identical HA and neuraminidase (NA) and thus display identical antigenic surfaces but have a distinct spherical or filamentous morphology. We established a correlative light and scanning electron microscopy (CLSEM) workflow to monitor the spread and morphology of virions at defined pulmonary or immunological pressures. We show that IAV infection in Calu-3 cells induces cell motility.

Furthermore, we discovered that spherical viruses exhibit increased entry kinetics and spread faster in diverse tissue cultures and at variable cell densities, as demonstrated in adherens junction deficient cells. Strikingly, our data show that neutralizing antibodies against HA are more effective in blocking spherical virions.

# Results

## Generation and characterization of IAV reporter viruses with predominant spherical or filamentous morphology

To exclusively compare IAV morphology-dependent effects, we generated two reporter viruses with distinct phenotypes, using a reverse genetics (RG) system (Hoffmann et al, 2000) with genetically modified plasmids in an influenza A/WSN/33 (H1N1) (WSN) background. We used a plasmid encoding mScarlet, a fluorescent protein, fused to the PA gene to generate WSN:PAmScarlet (Fig. 1A), based on previously published work (Tran et al, 2013). The mScarlet gene was codon-optimized to remove CpG dinucleotides to evade recognition by Zinc Finger Antiviral Protein (Ficarelli et al, 2019) and thereby increase the stability of the reporter viruses. As WSN is a lab-adapted strain with spherical morphology, we exchanged the WSN-M1 gene segment with the M1 segment of A/Udorn/307/72 (H3N2) to produce WSN-M1$_{Udorn}$:PAmScarlet with a predominantly filamentous morphology (Fig. 1B,E), as previously published (Bourmakina and Garcia-Sastre, 2005; Vahey and Fletcher, 2019b). M1$_{Udorn}$ contains 5 amino acid substitutions when compared to M1 from WSN (Fig. 1C). Cryo-electron microscopy (cryo-EM) analysis of viruses confirmed that WSN-M1$_{Udorn}$:PAmScarlet contained 79.34% ($n = 219$) of filamentous particles with a virion axis ratio >2 and a median particle length of 547 nm. WSN:PAmScarlet viruses were mainly spherical (80.50%, $n = 70$) (Fig. 1F). The median length of WSN:PAmScarlet virions was 132 nm. Overall, 34.25% of WSN-M1$_{Udorn}$ virions were longer than 1 µm compared to 2.86% for WSN. The maximum observed virion lengths were 4.78 µm for WSN-M1$_{Udorn}$ and 1.51 µm for WSN. Interestingly, cryo-electron tomography (cryo-ET) of spherical virions revealed gaps and kinking of the M1 layer, presumably to accommodate vRNPs (Fig. 1D, yellow arrowhead) during budding (Wachsmuth-Melm et al, 2024). HA spikes densely cover most of the surface on both spherical and filamentous virions. In contrast, NA is asymmetrically distributed, predominantly localized at one end of the particles (Calder et al, 2010) (Fig. 1D,E, white arrowheads). To validate that the reporter viruses carried the PA:mScarlet gene, we used fluorescence microscopy to image plaques formed in MDCK cells infected with WSN-M1$_{Udorn}$:PAmScarlet or WSN:PAmScarlet (Fig. EV1A,B). This showed that 71% of WSN-M1$_{Udorn}$:PAmScarlet and 42% of WSN:PAmScarlet rescued viruses expressed mScarlet (Fig. EV1C). The remaining fraction (29% and 58%) of viruses presumably eliminated the mScarlet. Note that non-fluorescent plaques were larger than fluorescent ones at 18 and 36 hpi, which indicates that mScarlet fused to PA yields a disadvantage to virus replication (Fig. EV1E).

## Quantification of fluorescent plaques demonstrates delayed spread of filamentous IAV

To quantitatively compare the spread of WSN-M1$_{Udorn}$:PAmScarlet (filamentous) or WSN:PAmScarlet (spherical) viruses, we analyzed

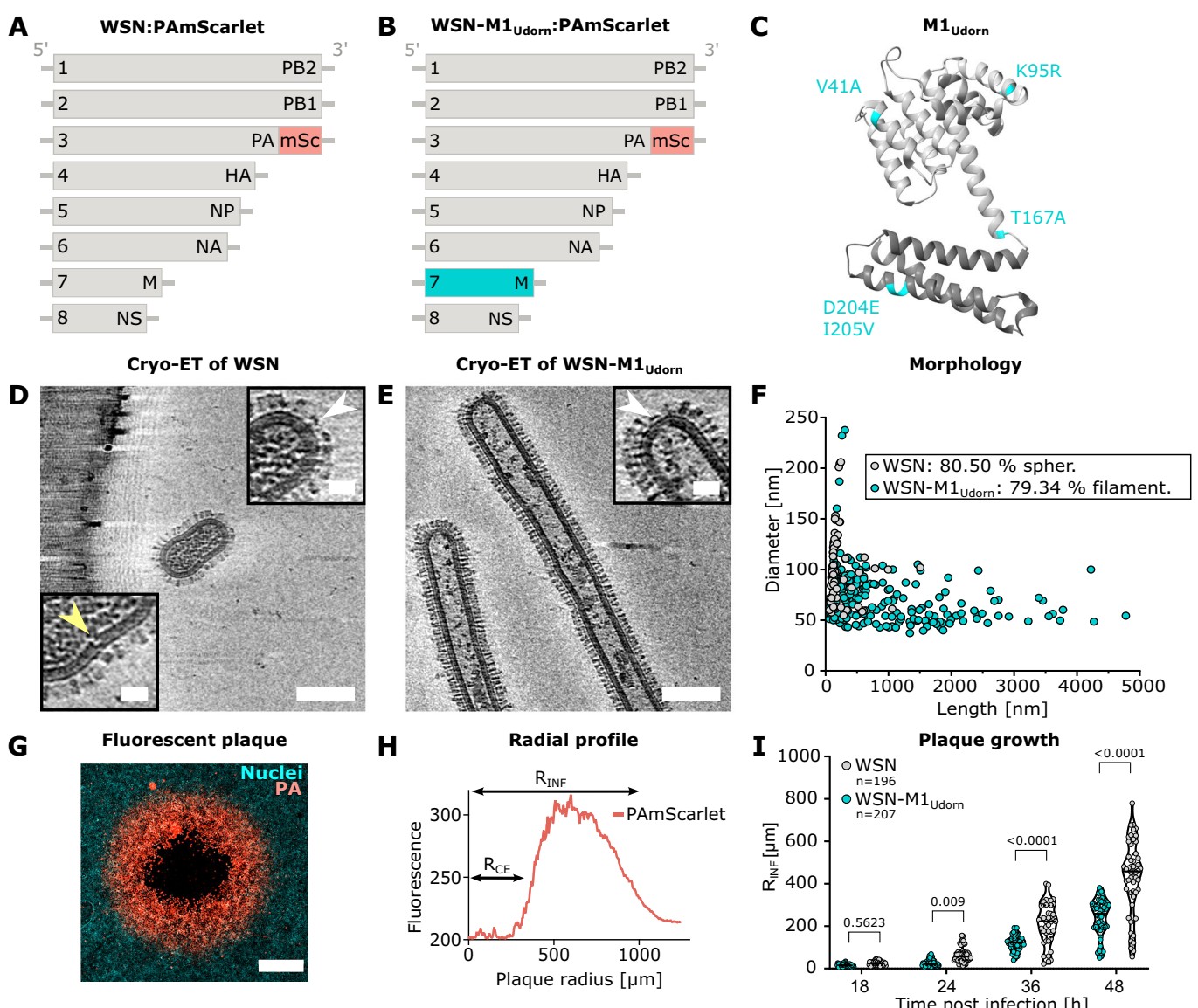

**Figure 1. Morphological characteristics and spread of spherical and filamentous reporter IAVs.**

(A) Schematic representation of the eight genomic segments of A/WSN/33:PAmScarlet (WSN:mScarlet) with the fluorescent reporter mScarlet (mSc, red) fused to the PA gene segment. (B) Genomic segments of WSN:PAmScarlet, where the segment 7 was exchanged by the M segment of A/Udorn/72 (cyan) (WSN-M1$_{Udorn}$:PAmScarlet). (C) Ribbon structure of the matrix protein 1 (M1) from A/WSN/33 (PDB: 6Z5L) harboring 5 amino acid substitutions (cyan) from M1 of A/Udorn/72. (D) Computed slices through a cryo-electron tomogram of an isolated WSN particle with a length/diameter ratio of 1.5. Scale bar: 100 nm. A gap in the M1 layer is indicated by a yellow arrowhead. The polarized distribution of neuraminidase at one end of the virion is indicated by a white arrowhead. Scale bar of zoom-ins: 20 nm. (E) Slice through cryo-electron tomogram of two isolated filamentous WSN-M1$_{Udorn}$ virions. Scale bar: 100 nm. Neuraminidase is indicated by a white arrowhead. Scale bar of zoom-in: 20 nm. (F) Quantification of virion diameters and lengths for WSN (gray) ($n = 70$) and WSN-M1$_{Udorn}$ (cyan) ($n = 219$) from TEM overview maps. Percentages of spherical (spher.: length/diameter ratio ≤2) and filamentous (filament.: length/diameter ratio >2) phenotypes are indicated. (G) Exemplary fluorescence image of a plaque in a MDCK cell monolayer (cyan), initiated by the infection with a single virus particle and spread of released viruses from the center outward. The plaque can be divided into three zones from the center: empty zone caused by cytopathic effect (black), surrounded by an infected cell zone showing WSN:PAmScarlet signal (red), and uninfected cells (cyan). Scale bar 500 μm. The panel is a zoom-in of Fig. EV1A. (H) Quantification of viral growth dynamics by radial profile analysis of the cytopathic effect radius ($R_{CE}$) from plaque center to the edge of empty zone and radius of infected cells ($R_{INF}$) from plaque center to the outer edge of PAmScarlet positive cells. (I) $R_{INF}$ of MDCK cells infected with WSN (gray) or WSN-M1$_{Udorn}$ (cyan) at different time points (18, 24, 36, 48 h) post infection. The plaque growth for WSN and WSN-M1$_{Udorn}$ infected cells were compared by two-way ANOVA followed by Tukey's multiple comparisons test. Exact *p* values: 5.623e-1, 8.992e-3, 2.652e-8, below 1.0e-15. Source data are available online for this figure.

each plaque using radial profile averaging of mScarlet fluorescence. We defined three different zones of infection representing detached cells (cytopathic effect), infected cells showing PAmScarlet signal, and uninfected cells (Figs. 1G and EV1D). The radial average of mScarlet signal showed a peak which allowed us to determine the radius of the cytopathic effect ($R_{CE}$) and the infection radius ($R_{INF}$) (Fig. 1H). The analysis of 403 plaques revealed that spherical viruses spread faster than filamentous viruses in MDCK cells (Fig. 1I).

## Virus morphology does not change throughout the course of infection

The assembly of filamentous virions requires a larger number of structural proteins and possibly also takes a longer time before the particle is released into the medium. This prompted us to analyze the morphology of budding virions during the infection and in particular, to address whether filamentous morphology is lost in the course of infection within a plaque. We established a CLSEM method (Fig. EV2) that can be applied to image virion morphology at different stages of infection spread within individual plaques. This method was first implemented on MDCK cells, a cell line often used for IAV propagation and plaque assays. This correlative approach enabled us to analyze the morphology of virions at the surface of infected cells at 36 and 42 hpi, after multiple rounds of infection had occurred (Fig. 2A–C,G–I). At the surface of MDCK cells infected with WSN:PAmScarlet, we observed a high number of spherical particles in proximity of the plaque center (Fig. 2E, yellow arrowhead), as well as at the border of the plaque (Fig. 2F, yellow arrowhead). On cells infected with WSN-M1$_{Udorn}$:PAmScarlet, filamentous viruses of several micrometers in length were observed (Fig. 2K,L, cyan arrowheads). Remarkably, both the spherical and filamentous virion morphologies were consistently preserved across all regions of the plaque (Fig. 2). Uninfected cells showed numerous filopodia, which have a wider and more variable diameter ($69 \pm 52$ nm, $n = 10$) than filamentous viruses (Fig. 2D,J–L). The data indicate that filamentous morphology is retained throughout multiple rounds of infection.

## Filamentous IAV exhibit delayed entry kinetics

Since our data show that the filamentous phenotype confers a disadvantage in cell-to-cell spread in cell culture, we next assessed the entry kinetics of both viruses using an entry uptake assay in A549 cells. The duration of viral entry can be determined by inhibiting endosomal acidification with $NH_4Cl$ at various time points. Interestingly, this unveiled that the uptake of filamentous viruses is considerably delayed with an entry half-time of 33 min in contrast to the spherical viruses whose uptake was at least twice as fast (Fig. 3A). In addition, we could show that filamentous viruses at the same multiplicity of infection (MOI = 3) led to the infection of around 52% of cells, while spherical viruses could infect about 84% of cells (Fig. 3B). Inhibitors dynasore and 5-($N$-ethyl-$N$-isopropyl)amiloride (EIPA), which target CME and macropinocytosis, respectively, effectively blocked the infection by both spherical and filamentous viruses (Fig. 3C,D). However, morphology-selective inhibition was not detected. Overall, our data showed that spread and entry kinetics of filamentous viruses

are reduced, indicating that the spherical morphology is better adapted to cell culture systems.

## Cryo-ET reveals intact filamentous viruses inside endosomal compartments

Given the delayed entry of filamentous IAV, we sought to visualize viral uptake and structurally characterize spherical and filamentous virions inside late endosomes using cellular cryo-ET. To this end, A549 cells were grown on EM grids and infected using a synchronized infection as done in our previous study (Klein et al, 2023). Infected cells were vitrified and milled using a focused ion beam to generate electron-transparent lamellae with a thickness of about 200 nm. Consistent with the entry kinetics assay showing the effective uptake of spherical viruses, cryo-ET analysis of cells infected with the spherical virus showed a high number of virions inside the endosomal compartment (Fig. 3E,I). Interestingly, spherical virions showed distinct axis ratios based on the presence of the M1 layer (Fig. EV3A,B). While virions with an intact M1 layer showed an ovoidal shape with an axis ratio above 1, virions with a disassembled M1 layer were spherical (Figs. 3F,J and EV3A,B). This indicates that upon acidification, M1 layer disassembly leads to shape relaxation to a more spherical shape. We were able to find one endosome containing three long filamentous viruses (length >500 nm) within a cell infected with WSN-M1$_{Udorn}$ (Fig. 3G,H). Although 3D segmentation of the filamentous virions showed that they were bent inside the endosome, the particles were intact (Fig. 3K,L).

## IAV spread in the absence of cell adherens junctions and in the presence of mucin

Since filamentous viruses can be several micrometers long, we hypothesized that they could confer an advantage in cell-to-cell spread when the contact between cells is disrupted. To test this hypothesis, we used MDCK-α-Catenin knock-out (KO) cells (Ollech et al, 2020) that show deficient adherens junctions and do not form a monolayer, as visible by plasma membrane staining (Fig. 4A). Impaired cell-cell contacts in uninfected MDCK-α-Catenin-KO cells are reflected by a decreased cell confluency at 24 and 48 h as compared to MDCK-WT cells (Fig. 4B). However, even at lower cell densities, IAV with spherical morphology exhibits an increased cell-to-cell spread compared to the filamentous virus (Fig. 4C). We next tested mucin, a key component of the mucus layer that serves as a pulmonary barrier against IAV infection. Sialic acids on mucins interact with HA but can be cleaved by NA, enabling viral penetration (Cohen et al, 2013; Kaler et al, 2022; Matrosovich et al, 2004). To evaluate the inhibitory effect, mucin was mixed into an agarose overlay at a concentration range of 0.5–2% and analyzed by fluorescent plaque assays in MDCK cells. Given that the inhibitory effect of mucin could also be due to a change in the viscosity of the agarose gel, we first evaluated the physical properties of the mucin-agarose mixtures. Our data revealed that the viscosity and elasticity of the mixture does not strongly depend on mucin concentrations (Fig. EV4A). Radial profile averaging of fluorescent plaques revealed that mucin effectively inhibits IAV spread (Fig. EV4B). However, mucin did not show an IAV morphology-dependent inhibitory effect as

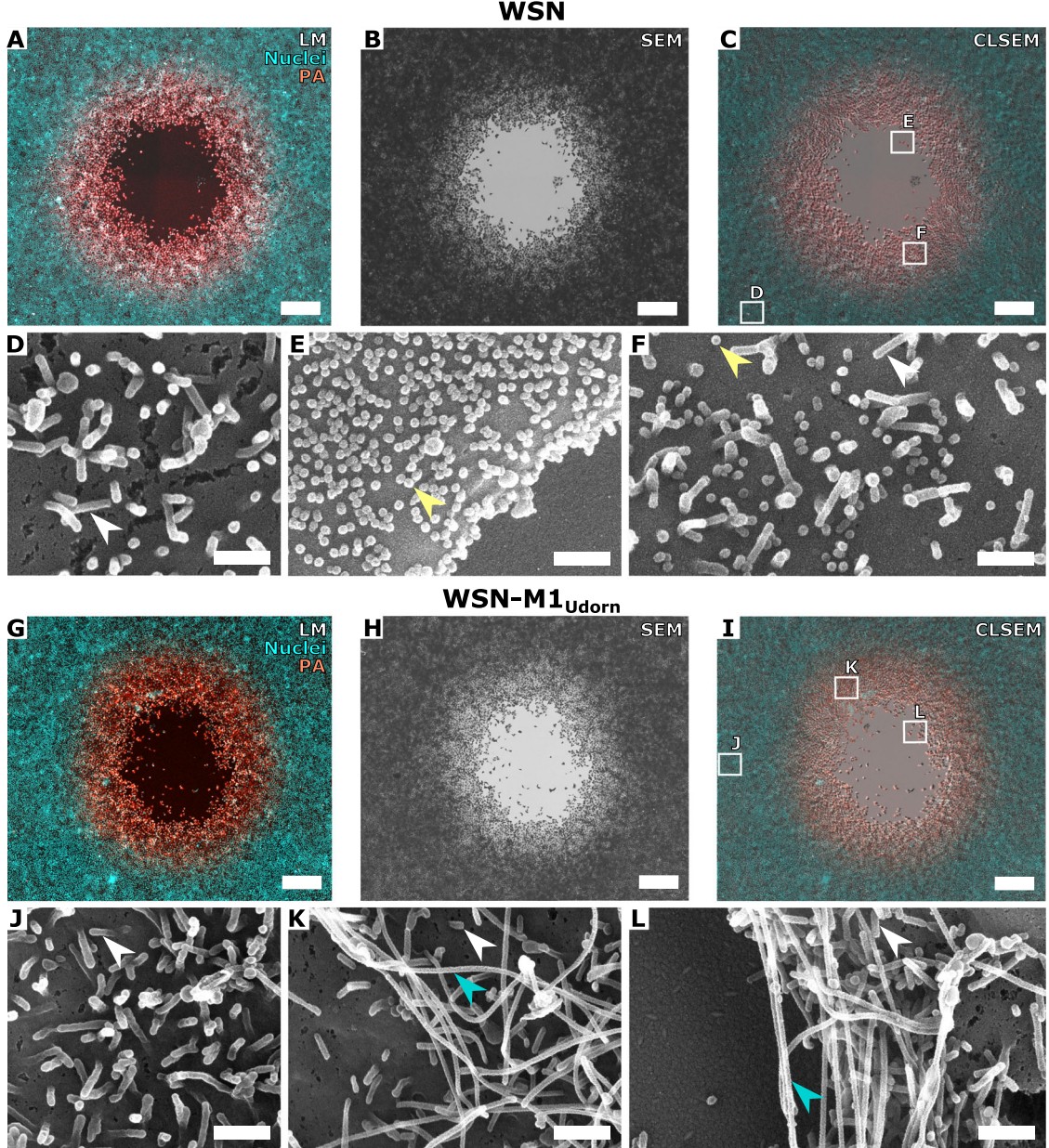

**Figure 2. Correlative light and scanning electron microscopy (CLSEM) of IAV cell-to-cell spread in MDCK cells.**

(**A**) Light microscopy image of a fluorescent plaque in a MDCK cell monolayer infected with spherical WSN:PA-mScarlet at 36 hpi, showing PA-mScarlet (red) and cell nuclei (cyan), scale bar: 100 µm. (**B**) Scanning electron microscopy (SEM) image of the plaque shown in (**A**), scale bar: 100 µm. (**C**) Correlation (CLSEM) of (**A, B**). (**D**) SEM image of an uninfected MDCK cell zone in proximity of the plaque in (**C**), scale bar: 0.5 µm. (**E**) SEM image from the plaque center of WSN:PA-mScarlet-infected MDCK cells, showing spherical IAV (yellow arrowhead), scale bar: 0.5 µm. (**F**) SEM image from the plaque periphery of WSN:PA-mScarlet-infected MDCK cells, showing spherical IAV (yellow arrowhead), scale bar: 0.5 µm. (**G**) Light microscopy image of a fluorescent plaque in a MDCK cell monolayer infected with filamentous WSN-M1$_{Udorn}$:PA-mScarlet at 42 hpi, showing PA-mScarlet (red) and cell nuclei (cyan). scale bar: 100 µm. (**H**) SEM image of the plaque shown in (**G**). Scale bar: 100 µm. (**I**) CLSEM of (**G, H**), scale bar: 100 µm. (**J**) SEM image of an uninfected MDCK cell zone in proximity of the plaque in (**I**), scale bar: 0.5 µm. (**K**) SEM image from the plaque center of WSN-M1$_{Udorn}$:PA-mScarlet-infected MDCK cells, showing filamentous IAV (cyan arrowhead) and filopodia (white arrowhead), scale bar: 0.5 µm. (**L**) SEM image from the plaque periphery of WSN-M1$_{Udorn}$:PA-mScarlet-infected MDCK cells, showing filamentous IAV (cyan arrowhead) and filopodia (white arrowhead). Scale bar: 0.5 µm. Source data are available online for this figure.

indicated by the relative plaque size decrease of WSN-M1$_{Udorn}$ compared to WSN across the mucin concentrations (Fig. 4E).

Previous studies have shown that NA plays a role in IAV mucus penetration (Vahey and Fletcher, 2019a). To compare the role of NA in the infection spread of spherical and filamentous IAV, we assessed the impact of the NA inhibitor zanamivir on viral spread (Hayden et al, 1997; Waghorn and Goa, 1998). Analysis of plaque growth showed that zanamivir had a comparable dose-dependent inhibitory effect on both viral morphologies (Figs. 4F and EV4C).

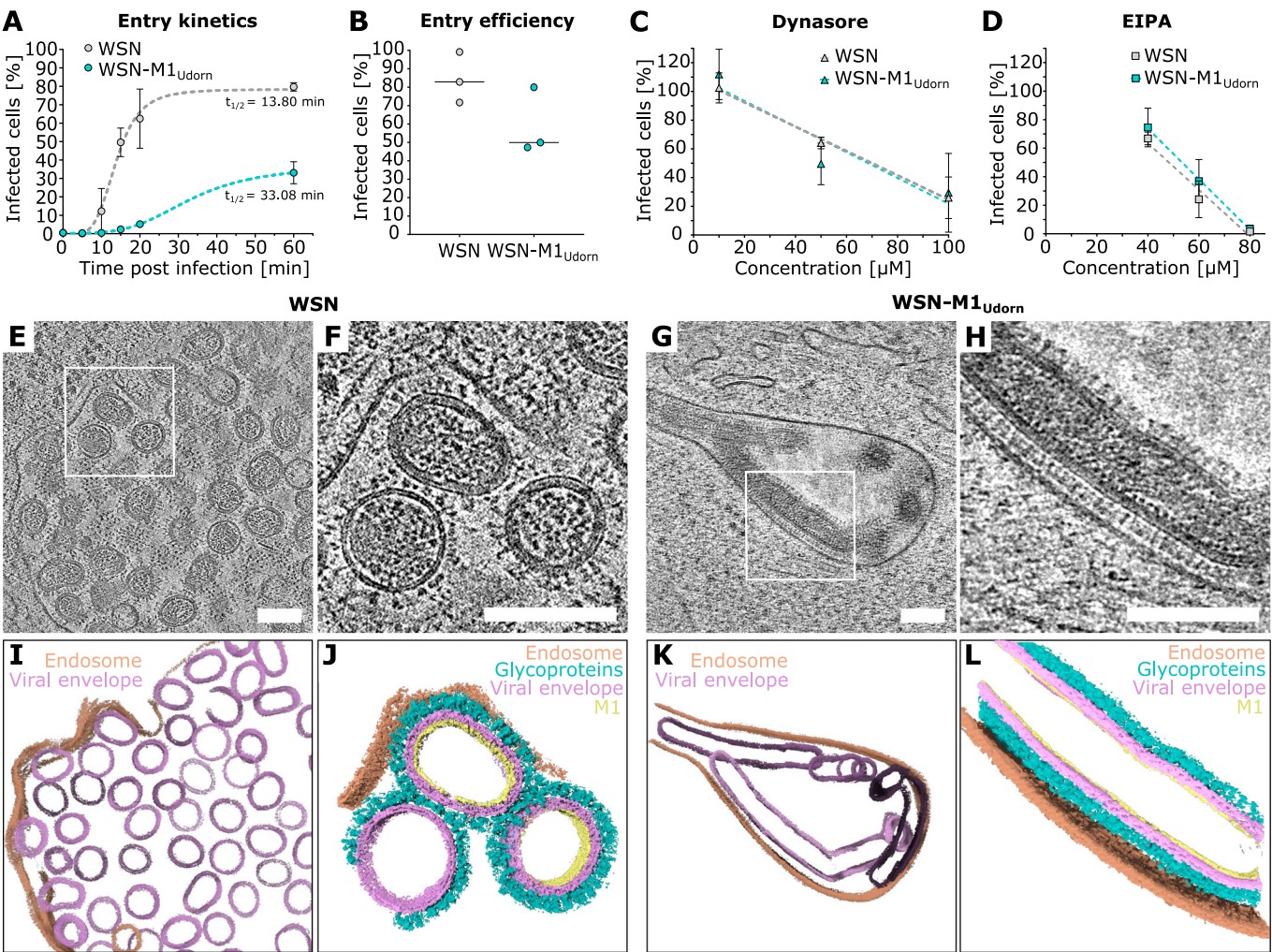

**Figure 3. Temporal and structural analysis of spherical and filamentous IAV cell entry.**

(A) Entry time course of WSN (gray) and WSN-M1$_{Udorn}$ (cyan) assessed by infection of A549 cells (MOI = 3), treatment with NH$_4$Cl (50 mM) at different time points (0, 5, 10, 15, 20, 60 min) post infection and fixation 12 h after the last time point. Infected cells were quantified by fluorescence microscopy and a penetration half-time was determined based on a four-parameter logistic (4PL) curve in three independent experiments for each virus strain: $R^2$(WSN) = 0.943, $t_{1/2}$(WSN) = 13.80 min, $R^2$(WSN-M1$_{Udorn}$) = 0.969, $t_{1/2}$(WSN-M1$_{Udorn}$) = 33.08 min. Means and standard deviations are indicated. (B) Percentage of A549 cells infected with WSN (gray) or WSN-M1$_{Udorn}$ (cyan) (MOI = 3, respectively) at 12 hpi determined by fluorescence microscopy in three independent experiments. Median(WSN) = 83.72, median(WSN-M1$_{Udorn}$) = 51.86. (C) Inhibitory effect of increasing dynasore concentrations on infection of A549 cells by WSN (gray) or WSN-M1$_{Udorn}$ (cyan) (MOI = 3 or 10), determined by fluorescence microscopy at 6 hpi. Cells were pre-treated with indicated dynasore concentrations for 1 h prior to infection. Values were normalized to DMSO-treated control. Means and standard deviations are indicated for three independent experiments. Linear regression fits: $R^2$(WSN) = 0.925, $R^2$(WSN-M1$_{Udorn}$) = 0.717. (D) Inhibitory effect of increasing EIPA concentrations on infection of A549 cells by WSN (gray) or WSN-M1$_{Udorn}$ (cyan) (MOI = 3 or 10), determined by fluorescence microscopy at 6 hpi. Cells were pre-treated with indicated EIPA concentrations for 1 h prior to infection. Values were normalized to DMSO-treated control. Means and standard deviations are indicated for three independent experiments. Linear fits: $R^2$(WSN) = 0.918, $R^2$(WSN-M1$_{Udorn}$) = 0.899. (E) Slices through a cryo-electron tomogram showing spherical WSN particles in an endosomal compartment of infected A549 cells at 15–30 min post infection, scale bar: 100 nm. (F) Zoom-in of three WSN virions from the highlighted region in E, scale bar: 100 nm. (G) Slices through a cryo-electron tomogram showing three filamentous WSN-M1$_{Udorn}$ particles in an endosomal compartment of an infected A549 cell at 15–30 min post infection, scale bar: 100 nm. (H) Zoom-in of one WSN-M1$_{Udorn}$ virion from the highlighted region in (G), scale bar 100 nm. (I–L) 3D segmentations of tomograms from (E–H). Color code: endosomes (orange), viral envelope (pink), viral glycoproteins (turquoise), matrix protein 1 (M1, yellow). Source data are available online for this figure.

Furthermore, to assess the infection spread of spherical and filamentous IAV in lung cells, we conducted fluorescent plaque assays in Calu-3 cells, which naturally produce mucins (Fig. 4D) (Lee et al, 2021). Interestingly, infection in Calu-3 cells did not result in plaques but in infection foci (Fig. 4G). Furthermore, our data show that filamentous viruses spread slower than spherical viruses also in Calu-3 cells (Fig. 4H), and that the morphology of spherical and filamentous viruses was retained throughout the infection foci (Fig. EV4D,E). Time-lapse imaging of Calu-3 foci revealed that cells infected by either WSN:PAmScarlet or WSN-M1$_{Udorn}$:PAmScarlet migrate towards the center of the infection focus (Fig. EV4G,K,L). The cell migration velocity, directionality and distance migrated towards the focus center were increased in IAV-infected Calu-3 cells as compared to uninfected cells (Fig. EV4G–M).

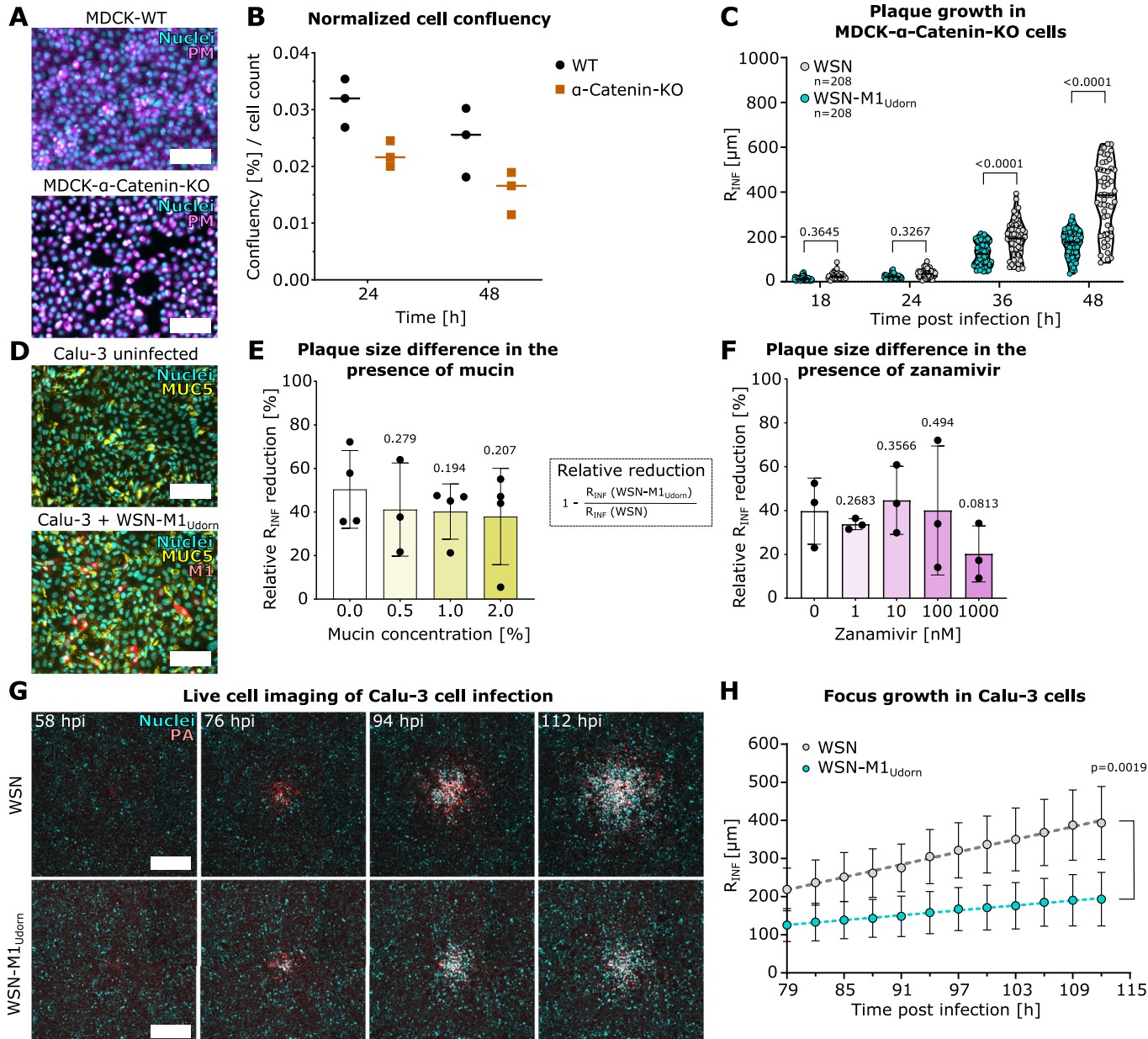

**Figure 4. Impact of cell-cell contact and mucin on IAV spread.**

(A) Representative images of MDCK-WT and MDCK-α-Catenin-KO cells after CellMask plasma membrane (PM) labeling (magenta) and nuclear counter stain (cyan), scale bars: 100 μm. (B) Cell densities of uninfected MDCK-WT cells (black) and MDCK-α-Catenin-KO cells (brown) as a ratio of area coverage / number of nuclei at 24 h and 48 h post addition of an avicel overlay. The mean of 3 biological replicates is indicated for each condition. (C) $R_{INF}$ of MDCK-α-Catenin-KO cells infected with WSN (gray) or WSN-M1$_{Udorn}$ (cyan) at different time points (18, 24, 36, 48 h) post infection. For each time point, the plaque growth for WSN and WSN-M1$_{Udorn}$ infected cells were compared by two-way ANOVA followed by Tukey's multiple comparisons test. Exact p values: 3.645e-1, 3.267e-1, 3.704e-6, below 1.0e-15. (D) Calu-3 cells uninfected (top) or infected with WSN-M1$_{Udorn}$ (bottom) after immunofluorescence staining of mucin 5AC (yellow), M1 (red) and nuclear staining (cyan), scale bars: 100 μm. (E) $R_{INF}$ reduction of WSN-M1$_{Udorn}$ plaques relative to WSN plaques in MDCK cells in the presence of indicated mucin concentrations (yellow). Relative reduction = 1 − (Radius of WSN-M1$_{Udorn}$/Radius of WSN × 100). Mean and standard deviations are indicated for four independent experiments. P values were calculated using one-sided Student's t test comparing each concentration with the untreated control. (F) $R_{INF}$ reduction of WSN-M1$_{Udorn}$ plaques relative to WSN plaques in MDCK cells in the presence of indicated zanamivir concentrations (magenta). Relative reduction and statistical significance were calculated as described in (E). Mean and standard deviations are indicated for three independent experiments. (G) Exemplary fluorescence microscopy images of WSN:PAmScarlet and WSN-M1$_{Udorn}$:PAmScarlet plaques from live cell imaging of Calu-3 cells at indicated time points with cell nuclei shown in cyan and PAmScarlet in red, scale bars: 500 μm. (H) $R_{INF}$ determined by live cell imaging of Calu-3 cells infected with WSN:PAmScarlet (gray) or WSN-M1$_{Udorn}$:PAmScarlet (cyan) between 79 and 112 hpi. Each data point represents the mean of at least 16 foci from 2 independent experiments. Standard deviations are indicated. Linear fits are indicated: $R^2$(WSN-M1$_{Udorn}$) = 0.994, $R^2$(WSN-M1$_{Udorn}$) = 0.993. The p value for 112 hpi was determined by Mann–Whitney test. Source data are available online for this figure.

## IAV infection spread in the presence of HA-binding neutralizing antibody

Filamentous virions contain significantly more HA glycoproteins than spherical virions. Based on the length distribution (Fig. 1F) and HA-HA spacing (Chlanda et al, 2016), we estimated that filamentous virions have on average 3.5 times more HAs (Fig. 5A). This prompted us to investigate whether filamentous virions will be able to escape from neutralization by a stalk-binding antibody MEDI8852, which has been shown to limit transmission of pandemic IAV (Paules et al, 2017) and has an impact on IAV morphology (Partlow et al, 2025). While we could observe that neutralizing antibodies inhibit the cell entry and spread of IAV, a higher concentration of antibodies was needed to inhibit filamentous viruses compared to spherical viruses (Fig. 5B,C). Treatment with 2.5 and 5 nM MEDI8852 minimized differences in spread velocity between WSN and WSN-M1$_{Udorn}$ (Fig. EV5). Next, we assessed morphological changes in budding IAV in the presence of neutralizing MEDI8852 antibodies using CLSEM and analyzed released virions from supernatants after serial passaging using cryo-EM. While CLSEM of plaques (Fig. 5F–I) as well as serial passaging of IAV (Fig. 5D) in the presence of neutralizing MEDI8852 antibodies showed that the number of produced WSN and WSN-M1$_{Udorn}$ virions was reduced (Fig. 5D), viral morphologies remained consistent under antibody pressure (Fig. 5E–I). WSN virions remained mostly spherical (93.33% spherical at passage 1; 91.46% spherical at passage 5) (Fig. 5E). In contrast, WSN-M1$_{Udorn}$ remained predominantly filamentous (80.77% filamentous at passage 1; 72.55% filamentous at passage 5). The mean length/diameter of WSN-M1$_{Udorn}$ virions remained greater than 7 in the presence of antibody and after five passages, whereas for WSN, the ratio stayed consistently around 1.5 (Fig. 5E).

## Discussion

While the pleomorphic nature of IAV particles directly contributes to higher infectivity and transmissibility in vivo, spherical morphology is selected for in cell culture (Seladi-Schulman et al, 2013). Recent studies have characterized heterogeneous IAV particles on a structural (Calder et al, 2010; Dadonaite et al, 2016; Harris et al, 2006) and genetic level and identified residues in the M1 protein as essential determinants of virion morphology (Bourmakina and Garcia-Sastre, 2003, 2005). However, the functional role of filamentous virions in the infected host and factors favoring this phenotype remain to be identified. One reason for this gap is the lack of quantitative approaches that systematically compare different virion morphologies in various conditions. Here, we investigated morphology-linked differences in host cell entry and spread of IAV in vitro. By establishing a fluorescent reporter virus system of distinct spherical and filamentous phenotypes as described by Bourmakina, Garcia-Sastre (Bourmakina and Garcia-Sastre, 2005), Vahey and Fletcher (Vahey and Fletcher, 2020) but identical surface antigens, we were able to track infection spread over time. Notably, the rescued reporter viruses used in this study exhibit around 80% of the phenotype. These results are in line with previous studies showing that M1 is the determinant for IAV morphology (Badham and Rossman, 2016; Calder et al, 2010). Our data demonstrate a disadvantage of

filamentous virions in cell-to-cell spread in MDCK cells and an MDCK-α-Catenin knock-out cell line which shows increased cell-to-cell distance in the monolayer. Hence, our data indicate that filamentous viruses do not gain an advantage in cell-to-cell spread kinetics at low cell densities, as we initially hypothesized.

The delay in the spread of filamentous viruses can occur at different stages of viral replication, and likely entry is one of the major factors. Since both viruses have the same genetic background the differences in virus replication cycle kinetics presumably occur at entry or exit levels. We showed that the endosomal escape of filamentous viruses is delayed by 20 min compared to spherical viruses. Delayed early infection of filamentous viruses was also reported in a study using spherical A/Udorn/307/72 and filamentous A/Udorn/307/72 10 A variants (Sieczkarski and Whittaker, 2005). Consistent with existing evidence showing that the entry of both filamentous and spherical IAV is blocked by inhibitors targeting macropinocytosis (Rossman et al, 2012), our data show that the entry of spherical and filamentous viruses is inhibited by EIPA or by dynasore to a similar extent. Since EIPA inhibits the acidification of endosomes (Gekle et al, 2001; Rossman et al, 2012), entry inhibition of spherical viruses cannot be excluded. Moreover, it has been shown previously that both spherical and filamentous viruses can induce macropinocytosis (de Vries et al, 2011; Rossman et al, 2012). Hence, both spherical and filamentous viruses are likely to utilize multiple entry pathways.

It is well-established that filamentous viruses undergo disintegration upon low pH treatment in vitro (Rossman et al, 2012). Hence, we analyzed the structure of spherical and filamentous virions inside the endosomes by cellular cryo-ET of cryo-focused ion beam-milled infected cells. This allowed us to capture long filamentous virions inside endosomes, which were bent but did not undergo disintegration into smaller components and were longer than those reported previously (Rossman et al, 2012). However, this disintegration is pH dependent and long filamentous virions were most likely in early endosomes as the HA was in a prefusion conformation. Notably, the endosomal environment led to morphological changes of spherical virions, which upon M1 layer disassembly became rounder.

Hence, our findings indicate that the increased entry kinetics of spherical viruses is a factor which drives the morphological adaptation towards spherical virions in vitro. However, other factors likely contribute to this adaptation, such as increased assembly efficiency for spherical viruses which require fewer building blocks for assembly. Previous studies using video-microscopy imaging of the budding respiratory syncytial virus revealed an average speed of filament elongation of 110–250 nm/s (Bachi, 1988). Assuming a similar budding velocity in IAV, it is unlikely that budding and growth of long filamentous virions significantly contributes to the delay in the cell-to-cell spread of filamentous viruses. The accumulation of spherical virions observed by SEM on the cell surface indicates that the rate-limited step in budding is likely membrane scission rather than budding particle growth. However, our data show that there are approximately twice as many released spherical viruses when compared to filamentous viruses (Fig. 5D). Hence, infected cells likely have the metabolic capacity to produce more spherical IAV particles than filamentous.

Since filamentous IAV spread is slower in cell culture, we hypothesized that they undergo morphological adaptation towards a spherical morphology throughout infection. In order to assess

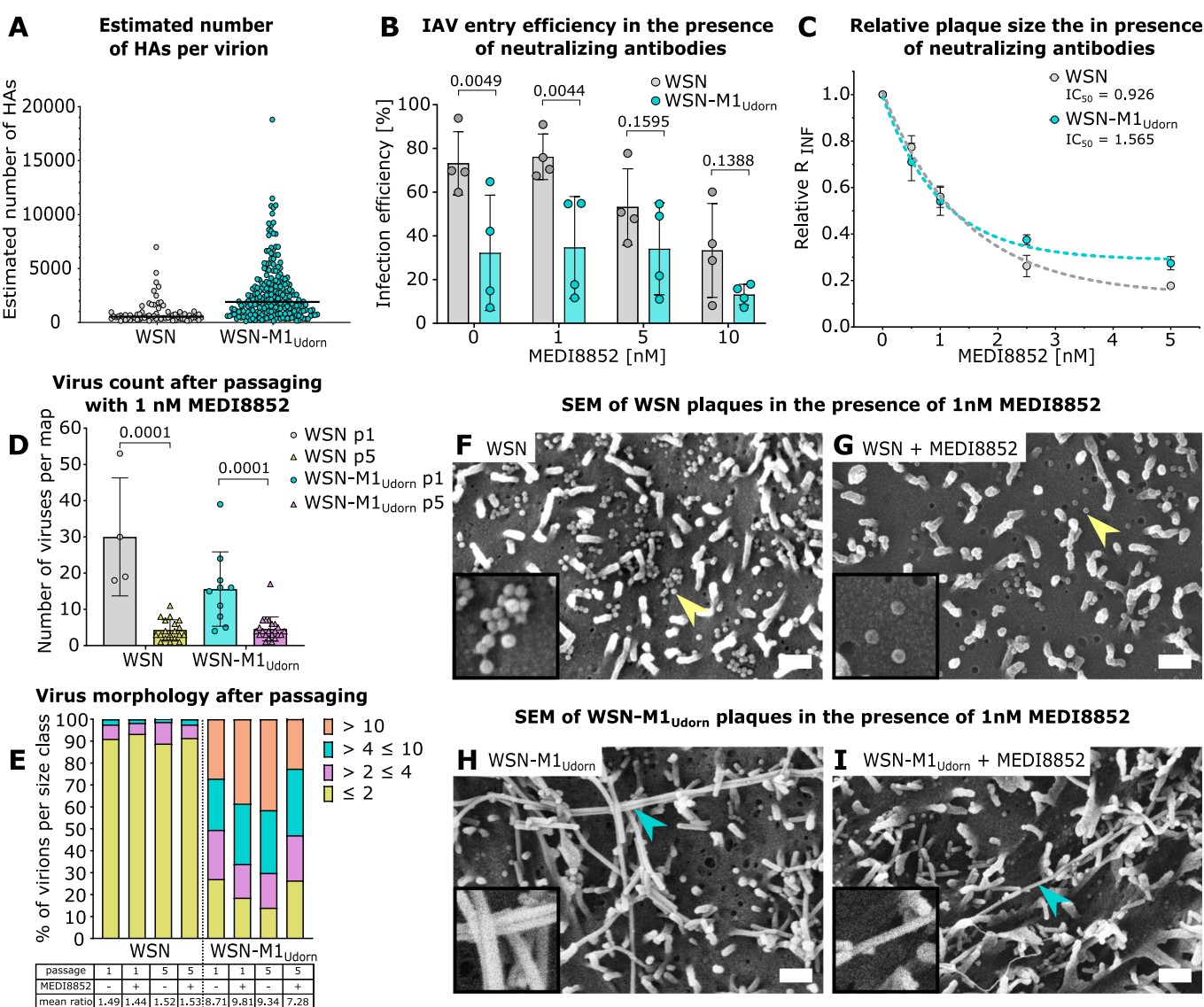

Figure 5. Effect of neutralizing anti-HA antibodies on morphology-dependent IAV spread.

(A) Estimated number of HAs per virion surface based on size measurements from Fig. 1F. Median(WSN) = 534.73, median(WSN-M1_Udorn) = 1904.31. The estimation was performed on 70 WSN and 219 WSN-M1_Udorn virions from one representative experiment. (B) Percentage of A549 cells infected by WSN (gray) or WSN-M1_Udorn (cyan) (MOI = 3, respectively) in the presence of 0, 1, 5 or 10 nM MEDI8852 antibody at 4 hpi. Mean values from four biological replicates with standard deviations are shown. Statistical analysis was done by two-way ANOVA. (C) Relative infection radius ($R_{INF}$) in MDCK cells infected with WSN (gray, $n = 388$) or WSN-M1_Udorn (cyan, $n = 921$) in the presence of 0.5, 1, 2.5 or 5 nM MEDI8852 at 36 hpi, normalized to untreated controls. Mean values and standard deviations from three independent experiments are indicated. $IC_{50}$ was determined from three-parameter logistic (3PL) curve fits: $R^2$(WSN) = 0.926, $IC_{50}$(WSN) = 0.926 nM, $R^2$(WSN-M1_Udorn) = 0.974, $IC_{50}$(WSN-M1_Udorn) = 1.565 nM. (D) Quantification of WSN (gray) and WSN-M1_Udorn (cyan) viruses per cryo-TEM map (612 μm²) after one (p1, circles) or five (p5, triangles) passages in MDCK cells in the presence of 1 nM MEDI8852 antibody. A minimum of 81 particles was analyzed per condition. Each data point represents the number of maps with mean values and standard deviations shown. Statistical significance was determined using an unpaired $t$ test. (E) Percentage of virions in each size class (length/diameter): ≤2 (yellow), >2–≤4 (magenta), >4–≤10 (cyan), >10 (orange) from serial passaging (p1 and p5) of WSN and WSN-M1_Udorn in the absence or presence of 1 nM MEDI8852 antibody in MDCK cells. For each condition, mean virion length/diameter ratios and numbers of analyzed particles (n) are indicated below. (F) Scanning electron microscopy (SEM) image of a plaque in MDCK cells infected with WSN without neutralizing antibody. Yellow arrowheads indicate spherical viruses, scale bar: 3 μm. (G) SEM image of a plaque in MDCK cells infected with WSN in the presence of 1 nM MEDI8852, scale bar: 3 μm. (H) SEM image of a plaque in MDCK cells infected with WSN-M1_Udorn without neutralizing antibody. Cyan arrowheads indicate filamentous viruses, scale bar: 3 μm. (I) SEM image of a plaque in MDCK cells infected with WSN-M1_Udorn with 1 nM MEDI8852, scale bar: 3 μm. Source data are available online for this figure.

whether the filamentous morphology is lost during several rounds of infection in cell culture (Seladi-Schulman et al, 2013), we established a CLSEM approach to image budding virions within different zones of a plaque using SEM. It can be assumed that the majority of viruses on the cell surface captured by SEM are budding particles consistent with transmission electron microscopy studies showing large quantities of budding virions connected to plasma membrane by a budding neck (Sugita et al, 2011). Interestingly, our results show that the morphology of budding virions is not altered as infection progresses. This indicates that virus clone propagation within a single plaque might not undergo sufficient replication cycles for morphology adaptation to occur. Alternatively, morphology maintenance or changes driven by adaptation may require a selective pressure that is lacking in vitro.

To investigate different types of selective pressure, we examined the role of mucin, NA inhibitor zanamivir and broadly neutralizing anti-HA stalk MEDI8852 antibodies. Mucins limit IAV infection in vivo (McAuley et al, 2017) and are known to be induced upon interferon treatment or IAV infection, and to effectively reduce the infection of different influenza A viruses (Iverson et al, 2022). Our data confirmed that the addition of mucin into the agarose overlay inhibits IAV infection in a dose-dependent manner. However, we did not observe a change in relative difference in spread between IAV with filamentous and spherical morphology in the absence or presence of mucin. Notably, we used porcine gastric mucin, which might differ structurally and in the sialic acid linkage types compared to human mucins (Nordman et al, 2002; Zhang et al, 2021). However, both in the porcine stomach and human airway, MUC5AC molecules are the predominant gel-forming mucins (Lillehoj et al, 2013). To assess the virus morphology-dependent spread in lung-derived mucins, we used Calu-3 cells, which produce MUC5AC mucin (Fig. 4D). In line with artificial mucin addition, the spread of spherical IAV in mucin-producing Calu-3 cells was improved compared to filamentous IAV virions (Fig. 4H). Studies have demonstrated that culturing Calu-3 cells at the air-liquid interface enhances their differentiation and mucus production compared to a liquid-covered culture (Grainger et al, 2006). Hence, morphology-dependent spread remains to be characterized in polarized cells cultured at the air–liquid interface. Unlike in MDCK cells, the infection of Calu-3 cells resulted in the formation of infection foci rather than plaques. Surprisingly, time-lapse movies revealed that infected Calu-3 cells move faster than uninfected cells towards the center of foci. While this needs to be further studied, the data indicate that Calu-3 cells are migrating towards the center to replace the dying cells. In addition, these results suggest that IAV infection triggers cell motility, which was reported, for example, in vaccinia virus-infected cells (Valderrama et al, 2006).

Importantly, we could show that the presence of neutralizing anti-HA stalk antibodies leads to the loss of the infection spread advantage of spherical IAV. This finding is in agreement with previous evidence showing that neutralizing antibodies have a stronger effect on spherical IAV (Li et al, 2021), and suggests that filamentous viruses can better escape neutralization by antibodies due to a much higher number of HAs. This is consistent with a recent work that used flow virometry (Zamora and Aguilar, 2018), showing that filament assembly better withstands the antibody pressure (Partlow et al, 2025). However, cryo-EM analysis of released viruses did not reveal any dramatic shape changes after passaging in the presence of an anti-HA stalk antibody. Furthermore, flow virometry is limited to released virions and does not take into account budding virions. CLSEM analysis of budding viruses confirmed that virus morphology does not dramatically change during infection of MDCK or Calu-3 cells. While flow virometry offers a high throughput, electron microscopy provides more precise morphology measurements at the single-virion level and can unambiguously distinguish impurities and viral aggregation.

Given that IAV release is dependent on NA activity and filamentous virions exhibit a higher HA:NA ratio, it was hypothesized that filamentous virions would be more susceptible to neuraminidase inhibitors. However, treatment with zanamivir resulted in comparable inhibition of viral spread between spherical and filamentous IAV populations (Fig. 4F).

Overall, we provide an imaging method combining fluorescence and scanning electron microscopy to monitor virus spread and morphology in situ. Our findings suggest that morphology-dependent IAV entry kinetics affect spread. Our data provide further evidence that spherical IAV exhibit accelerated infection spread through faster entry kinetics and greater entry efficiency. In contrast, filamentous IAV morphology is advantageous in the presence of pressure exerted by pulmonary barriers such as neutralizing antibodies.

## Methods

**Reagents and tools table**

| Reagent/resource | Reference or source | Identifier or catalog number |
|---|---|---|
| **Experimental models** | | |
| A549 cells (*H. sapiens*) | (Giard et al, 1973), ATCC | CCL-185 |
| Calu-3 cells (*H. sapiens*) | Prof. Ralf Bartenschlager, Heidelberg University, Germany | N/A |
| HEK 293T cells (*H. sapiens*) | Dr. Marco Binder, DKFZ, Heidelberg, Germany | N/A |
| Influenza A/WSN/1933 (WSN) | (Hoffmann et al, 2000) | N/A |
| Influenza A/WSN/1933 containing M1 from influenza A/Udorn/72 (WSN-M1$_{Udorn}$) | (Vahey and Fletcher, 2020) | N/A |
| MDCK cells (*Canis lupus familiaris*) | Dr. Maria João Amorim, Instituto Gulbenkian de Ciência, Oeiras, Portugal | N/A |
| MDCK-α-catenin-KO cells (*Canis lupus familiaris*) | Prof. Elisabetta Ada Cavalcanti-Adam, Max Planck Institute for Medical Research, Heidelberg, Germany | N/A |
| WSN:PAmScarlet | (Klein et al, 2023) | N/A |
| WSN-M1$_{Udorn}$:PAmScarlet | This study | N/A |
| **Recombinant DNA** | | |
| pcDNA3.1-M1-Udorn-M2-WSN | Dr. Michael Vahey, Washington University, St Louis, MO, USA and Prof. Daniel Fletcher, University of California Berkeley, Berkeley, CA, USA | N/A |
| pHW2000-PB2-WSN | Prof. Ervin Fodor, University of Oxford, UK | N/A |

| Reagent/resource | Reference or source | Identifier or catalog number |
|---|---|---|
| pHW2000-PB1-WSN | Prof. Ervin Fodor, University of Oxford, UK | N/A |
| pHW2000-PA-WSN-mScarlet codon optimized | (Klein et al, 2023) | N/A |
| pHW2000-HA-WSN | Prof. Ervin Fodor, University of Oxford, UK | N/A |
| pHW2000-NP-WSN | Prof. Ervin Fodor, University of Oxford, UK | N/A |
| pHW2000-NA-WSN | Prof. Ervin Fodor, University of Oxford, UK | N/A |
| pHW2000-M-WSN | Prof. Ervin Fodor, University of Oxford, UK | N/A |
| pHW2000-NS-WSN | Prof. Ervin Fodor, University of Oxford, UK | N/A |
| **Antibodies** | | |
| Goat anti-mouse IgG, Alexa Fluor 488 | ThermoFisher Scientific, Invitrogen | A11029 |
| MEDI8852, mouse IgG anti-HA | (Kallewaard et al, 2016; Partlow et al, 2025) | N/A |
| Mouse anti-M2 | Abcam | ab5416 |
| Mouse anti-NP | Merck, Sigma-Aldrich | MAB8257 |
| **Chemicals, enzymes and other reagents** | | |
| Agarose | Biozym | 840004 |
| Avicel microcrystalline cellulose | FMC Corporation | RC-581 |
| Bovine Serum Albumin (BSA) | Merck, Sigma-Aldrich | A7030 |
| CellMask | ThermoFisher Scientific, Invitrogen | C37608 |
| Crystal violet solution | Merck, Sigma-Aldrich | V5265 |
| DAPI | Merck, Sigma-Aldrich | D9542 |
| DMEM 2X | Merck, Sigma-Aldrich | SLM-202 |
| DMEM-F12-GlutaMAX | ThermoFisher Scientific, Gibco | 10565018 |
| DMEM-GlutaMAX-I | ThermoFisher Scientific, Gibco | 61965026 |
| DMSO | Merck, Sigma-Aldrich | D2650 |
| Dulbecco's Phosphate Buffered Saline (PBS) | Merck, Sigma Aldrich | D8537 |
| Dynasore | Merck, Sigma-Aldrich | 324410 |
| EGTA | Merck, Sigma-Aldrich | 03777 |
| EIPA | Merck, Sigma-Aldrich | A3085 |
| fetal bovine serum (FBS) | ThermoFisher Scientific, Gibco | 10270-106 |
| HEPES | ThermoFisher Scientific, Gibco | 15630080 |
| Hoechst 33342 | Merck, Sigma Aldrich | B2261 |
| Immunogold Protein A 10 nm | Aurion, Biotrend | 810.111 |
| $MGCL_2$ | Merck, Sigma-Aldrich | M2393 |
| Mucin from porcine stomach | Merck, Sigma-Aldrich | M1778 |
| Opti-MEM medium | ThermoFisher Scientific, Gibco | 31985062 |
| Paraformaldehyde 16% (PFA) | Electron Microscopy Sciences | 15710 |
| Penicillin/streptomycin (P/S) (10.000 units/ml) | ThermoFisher Scientific | 15140122 |
| PIPES | Merck, Sigma-Aldrich | P1851 |
| ProLong Glass Antifade Mountant | ThermoFisher Scientific, Invitrogen | P36982 |
| Protein A-coated colloidal gold (10 nm) | Aurion | 110.111 |
| Sodium pyruvate | ThermoFisher Scientific, Gibco | 11360070 |
| Sylgard 184 | DOW, Farnell | 101697 |
| TransIT-LT1 | Mirus Bio | MIR 2300 |
| Triton X-100 | Merck, Sigma-Aldrich | T8787 |
| Trypsin from bovine pancreas, TPCK treated (TPCK-Trypsin) | Sigma-Aldrich | T1426 |
| Tween 20 | Carl Roth GmbH + Co. KG | 9127.1 |
| Zanamivir | Merck, Sigma-Aldrich | SML0492 |
| **Software** | | |
| AreTomo | (Zheng et al, 2022) | N/A |
| Dragonfly | Comet Technologies Canada Inc. | N/A |
| ImageJ/Fiji | (Baggethun, 2009; Schindelin et al, 2012) | N/A |
| IMOD | (Kremer et al, 1996) | N/A |
| MAPS (3.3) | ThermoFisher Scientific | N/A |
| Motioncor2 | (Zheng et al, 2017) | N/A |
| Parallel Cryo-Electron Tomography (PACEtomo) | (Eisenstein et al, 2023) | N/A |
| Prism (10.1.2) | GraphPad | N/A |
| SerialEM | (Mastronarde, 2005) | N/A |
| TRIOS | TA Instruments | N/A |
| ZEN (blue edition) | ZEISS | N/A |
| **Other** | | |
| Aquilos 2 cryo-FIB | ThermoFisher Scientific | N/A |
| Celldiscoverer 7 microscope | ZEISS | N/A |
| EM ACE600 | Leica Microsystems | N/A |
| EM CPD300 | Leica Microsystems | N/A |
| EM GP2 plunge-freezing device | Leica | N/A |
| HR 20 Discovery Hybrid Rheometer | TA Instruments | N/A |
| K3 direct electron detector | Gatan | N/A |
| Titan Krios cryo-TEM | ThermoFisher Scientific | N/A |
| Whatman 1 | Cytiva | 1001-055 |
| Quanta Imaging Filter | Gatan | N/A |
| Quantifoil R2/1, holey carbon film, Cu 200 mesh | Quantifoil | N1-C15nCu 20-01 |
| Quantifoil R1.2/20, holey $SiO_2$, Au 200 mesh | Quantifoil | N1-S20nAu 20-01 |

## Methods and protocols

### Cell lines

HEK 293T (human embryonic kidney 293T) and MDCK cell lines were maintained in DMEM-GlutaMAX-I medium supplemented with 10% fetal bovine serum (FBS) and 1% penicillin/streptomycin (P/S). Calu-3 (human lung adenocarcinoma) cells were grown in DMEM-GlutaMAX-I medium supplemented with 20% FBS, 1% P/S, and 10 mM sodium pyruvate. A549 (human alveolar basal epithelial adenocarcinoma) cells were maintained in DMEM-F12-GlutaMAX supplemented with 10% FBS and 1% P/S. All cells were cultured at 37 °C in a humidified 5% $CO_2$ atmosphere and passaged twice a week in a 1:10 ratio or 1:3 (for Calu-3 cells). Cell lines were tested for Mycoplasma contamination every 3 months.

### Fluorescent influenza A reporter viruses

All work with infectious IAV was performed under biosafety level (BSL-)2 conditions at BioQuant or CIID (Heidelberg University, Germany), following approved operating procedures. Viruses used in this study were rescued by RG, using eight bidirectional plasmids as described by (Hoffmann et al, 2000). Two reporter virus strains with the genetic background of influenza A/WSN/33, carrying PA tagged with a codon optimized mScarlet, were produced based on (Bindels et al, 2017; Klein et al, 2023; Tran et al, 2013). Predominantly spherical WSN:PAmScarlet was rescued by transfecting the RG plasmids: pHW2000-PB2-WSN, pHW2000-PB1-WSN, pHW2000-PA-WSN-mScarlet, pHW2000-HA-WSN, pHW2000-NP-WSN, pHW2000-NA-WSN, pHW2000-M-WSN, pHW2000-NS-WSN. WSN-M1$_{Udorn}$:PAmScarlet was produced by replacing plasmid pHW2000-M-WSN with pcDNA3.1-M1-Udorn-M2-WSN. This plasmid contains the M1 sequence from influenza A/Udorn/72 with 5 amino acid substitutions as compared to M1 from A/WSN/33 and was previously shown to confer a predominantly filamentous virion phenotype (Vahey and Fletcher, 2020).

HEK 293T cells were seeded into 10 cm cell culture dishes at a density of $4 \times 10^6$ cells per dish and grown over night. The transfection mix was prepared by mixing 2 ml Opti-MEM medium with 2.5 μg of each of the 8 RG plasmids and 60 μl TransIT-LT1 transfection reagent and incubation at room temperature (RT) for 30 min. The mix was subsequently added dropwise to the cells followed by incubation for 24 h at 37 °C and 5% $CO_2$. The cell culture medium was replaced by FBS-free DMEM-GlutaMAX-I supplemented with 1% P/S, 0.3% BSA and 2 μg/μl TPCK-Trypsin. In all, $4 \times 10^6$ MDCK cells were seeded on the transfected HEK 293T cells for co-culture. Once a cytopathic effect was visible, cell supernatant was harvested and cleared from cell debris by centrifugation at $1000 \times g$ for 10 min. The virus-containing supernatant (P0) was snap-frozen in $LN_2$ and stored at −80 °C.

### Plaque assay for titer determination

MDCK cells were seeded into 6-well plates (Corning) at a density of $1 \times 10^6$ cells/well and grown into a monolayer overnight. Plaque assays were performed in duplicates. IAV aliquots stored at −80 °C were slowly thawed on ice for 1 h. A virus dilution series from $10^{-3}$ to $10^{-8}$ was prepared in 4 °C cold FBS-free DMEM-GlutaMAX-I medium. The cell monolayer was washed once with FBS-free DMEM-GlutaMAX-I medium to remove residual FBS. Next, 0.8 ml of the virus dilutions were added to the cells followed by incubation at 37 °C and 5% $CO_2$ to allow virus attachment and adsorption.

Unbound virus was aspirated, and cells were washed with PBS. Each well was overlayed with 3 ml microcrystalline cellulose overlay consisting of FBS-free DMEM 2X medium supplemented with 2% P/S, 50 mM HEPES, 7.4 g/l $NaHCO_3$, 0.3% BSA, and 2 μg/μl TPCK-Trypsin that was mixed with 2.4% Avicel at a 1:1 ratio. After incubation at 37 °C and 5% $CO_2$ for 48 h without moving the plate, cells were washed 3 times with PBS and fixed with 4% PFA in PBS for 30 min. Crystal violet staining (1% in $H_2O$) was performed for 10 min at RT followed by 3 washing steps with $H_2O$. Plaques were manually counted and the average virus titer ($T_{virus}$) from all dilutions was determined by

$$T_{virus} = \frac{\text{number of plaques}}{\text{dilution factor} * \text{inoculum volume [ml]}}$$

### Morphological characterization of viruses

To morphologically characterize the produced IAV strains by cryo-EM, virus-containing supernatants were thawed on ice for 1 h. In the meantime, EM grids (Quantifoil R2/1, holey carbon film, Cu 200 mesh) were glow discharged. Undiluted virus was mixed with 10 nm protein A-coated colloidal gold. In all, 3 μl virus were applied onto the grid prior to plunge-freezing. Plunge-freezing into liquid ethane was performed using an automatic EM GP2 plunge-freezing device under the following conditions: chamber temperature: 25 °C, humidity: 80%, back-side blotting: 2–3 s. Grids were stored in $LN_2$ until imaging. Cryo-TEM data was collected with SerialEM (Mastronarde, 2005) using a Titan Krios cryo-TEM operated at 300 keV and equipped with a Quanta Imaging Filter with an energy filter slit set to 20 eV and a K3 direct electron detector. First, grids were mapped at ×8700 magnification (pixel spacing: 10.64 Å). From these maps, the length and diameter of virions from outer membrane to outer membrane was measured in IMOD (Kremer et al, 1996). For filamentous virions, the diameter was measured at three locations and averaged. Measurements were plotted in Prism 10.1.2. At representative positions of the map, tilt series were acquired at ×33,000 magnification (pixel spacing: 2.67 Å) using Parallel Cryo-Electron Tomography (PACEtomo) (Eisenstein et al, 2023) in SerialEM (Mastronarde, 2005) with the following setup: dose-symmetric tilting scheme (Hagen et al, 2017), nominal tilt range from 60° to −60° and 3° increments, target defocus −3 μm, electron dose per record $3e^-/\text{Å}^2$. Drift correction of acquired movies was done with Motioncor2 (Zheng et al, 2017). Tomograms were reconstructed in IMOD using the following parameters: tilt series alignment with patch tracking, weighted back projection with simultaneous iterative reconstruction technique (SIRT)-like filter equivalent to five iterations, dose-weighting, 3D contrast transfer function (CTF) correction.

### Fluorescent plaque assays

For fluorescent plaque assays, MDCK, MDCK-α-catenin-KO, or Calu-3 cells were seeded at a density of $1 \times 10^6$ cells/well in 6-well plates and grown into a monolayer. IAV aliquots were thawed on ice for 1 h and diluted to 50–100 PFU/well in FBS-free DMEM-GlutaMAX-I medium. Cells were washed once with FBS-free DMEM-GlutaMAX-I before infection for 1 h at 37 °C and 5% $CO_2$. The monolayer was washed 2 times with PBS and overlayed with avicel overlay as described above. At 18, 24, 36 or 48 hpi, the overlay was removed, cells were washed 3 times with PBS and chemically fixed with 4% PFA in PBS for 30 min at RT.

For additional immunofluorescence staining of IAV-NP or M2, cells were washed three times with PBS and subsequently permeabilized with 0.2% Triton X-100 in PBS for 5 min at RT. Next, cells were washed 3 times with PBS for 5 min and blocking was performed for 30–60 min with 3% BSA in PBS-T (PBS supplemented with 0.1% Tween-20). After one washing step with dilution buffer (1% BSA in PBS-T), cells were incubated for 1 h at RT with mouse anti-NP or mouse anti-M2 primary antibodies diluted 1:500 in dilution buffer. Cells were then washed 3 times for 5 min with PBS-T before adding the secondary goat anti-mouse Alexa Fluor 488 antibody and DAPI, both at a dilution of 1:1000 in PBS-T. Plates were incubated for 1 h at RT in the dark and subsequently washed 3 times with PBS.

Fluorescence microscopy data was acquired using the ×5 objective lens of the Zeiss CellDiscoverer 7 microscope equipped with an Axiocam 712 camera. Tile sets covering 80% of the well were acquired in three channels with LED excitation of DAPI at 385 nm, Alexa Fluor 488 at 470 nm and mScarlet at 657 nm. Tiles were stitched in the ZEN software.

### Analysis of IAV spread

Virus spread of spherical and filamentous IAV was assessed by radial density profile measurements in ImageJ/Fiji. For each plaque, a circle was drawn above the center of each plaque and the fluorescent signal of PA-mScarlet, NP or M2 was radially averaged. The cytopathic effect radius ($R_{CE}$) represents the distance from the center of the plaque and the inner edge of infected cells (Fig. EV1D). The infection radius ($R_{INF}$) was defined as the distance between the center of the plaque and the outer edge of infected cells. These parameters were calculated for different time points after infection. Individual parameters and regression equations were plotted in Prism 10.1.2. The reduction of $R_{INF}$ from WSN-M1$_{Udorn}$ plaques relative to WSN plaques was calculated using the following formula: Relative reduction [%] = 100 − (Mean radius of WSN-M1$_{Udorn}$ plaques/Mean radius of WSN plaques × 100). To assess the reporter efficiency, growth dynamics ($R_{INF}$) of plaques resulting from spherical WSN and reporter WSN:PAmScarlet viruses were determined from M2 signals of WSN and PA-mScarlet signals of WSN:PAmScarlet for 10 plaques at 18, 24, and 36 hpi, respectively (Fig. EV1C,E).

### CLSEM of plaques

For SEM, MDCK cells were grown on indium tin oxide (ITO)-coated coverslips placed in 6-well plates until confluent and infected with fluorescent reporter IAV as described above (Fig. EV2). For plaque assays in the presence of neutralizing antibodies, 1 nM MEDI8852 antibody was added to the overlay medium. After chemical fixation, DAPI staining was performed at a 1:1000 dilution for 5 min. Plaques were imaged using the ×5 objective of the Zeiss CellDiscoverer 7 microscope. After fluorescent image acquisition, cover slips were washed with PHEM (15 mM PIPES, 6.25 mM HEPES, 2.5 mM EGTA, 0.5 mM MgCl$_2$) or 0.1 M cacodylate buffer. Next, cells were incubated with 1% OsO$_4$ in cacodylate or PHEM buffer at 4 °C for 30 min, followed by washing with cacodylate or PHEM buffer (Fig. EV2). Dehydration was achieved by addition of increasing concentrations of acetone (25%, 50%, 75%, 95%, 100%) to the sample and incubation for 10 min, respectively. Lastly, critical point drying was done on an EM CPD300 at 17 °C and 63.5 bar and the sample was sputter coated with a 5 nm thick layer of Au/Pd (80/20) using the EM ACE600. Samples were mapped by SEM using an Aquilos 2 dual-beam cryo-focused ion beam-scanning electron microscope at magnifications between ×100 and ×50,000 using 5 keV and 0.1 nA. Correlation with LM images was performed in the MAPS 3.3 software. The cell topology and virus morphologies were quantified with ImageJ/Fiji. For each condition, virus counts were determined from at least 4 images at ×50,000 magnification.

### Estimation of HA count per virion

To estimate the number of HA proteins per virion, the virion surface area was calculated based on particle length and diameter measurements from cryo-EM maps described above. For spherical IAV, we assumed perfect spherical symmetry using the formula $A = \pi \times d^2$. The basis for filamentous virion surface area is a cylinder with $A = 2\pi r^2 + 2\pi r l$. Subsequently, the number of HAs per virion was calculated by assuming 7 proteins per 500 nm$^2$ as described in Chlanda et al (2016).

### Cell densities

To determine cell densities, MDCK and MDCK-α-Catenin-KO cells were seeded into 6-well plates at a density of $1 \times 10^6$ cells/well and grown for 24 h. Cells were overlaid with 2.4% avicel in FBS-free DMEM-GlutaMAX-I and fixed with 4% PFA at 24 or 48 h after addition of the overlay. Cells were stained with DAPI and CellMask plasma membrane stain at a 1:1000 dilution in PBS for 10 min at RT. Fluorescent images were acquired at Zeiss CellDiscoverer 7 using the ×5 objective. Cell numbers were obtained by the segmentation of cell nuclei using the StarDist plugin in ImageJ/Fiji. The confluency was calculated by applying a threshold in ImageJ/Fiji and measuring the cell-covered area from the plasma membrane stain. Cell densities were represented as the ratio of cell-covered area to the number of cells for three independent replicates.

### Plaque assay in the presence of mucins

Dose-dependent effects of mucins on IAV cell-to-cell spread w4ere studied by plaque assay in the presence of mucin from porcine stomach type III in a concentration range of 0–2% (w/v), consistent with physiological conditions (Ridley and Thornton, 2018) and technical limitations. A 5% (w/v) mucin stock solution was prepared by dissolving mucin in DMEM 2× by stirring at 37 °C for 1 h. The stock was then UV-inactivated for 15 min to avoid microbial contamination and further diluted to 2%, 1% and 0.5% in DMEM 2×. The mixtures were supplemented with 2% P/S, 50 mM HEPES, 0.6% BSA and 4 μg/μl TPCK-Trypsin. A confluent MDCK cell monolayer in 6-well plates was infected with a dilution of WSN:PAmScarlet or WSN-M1$_{Udorn}$:PAmScarlet as described earlier. Meanwhile, mucin dilutions were mixed with 1% (w/v) agarose heated to 56 °C. Infected cells were overlaid with the mucin-agarose hydrogel and incubated at 37 °C and 5% CO$_2$ for 60 h. Immunofluorescent staining of NP was performed and fluorescent plaques were analyzed according to the above-mentioned protocol.

### Physical properties of mucin-agarose hydrogels

Rheological properties of mucin-agarose hydrogels were performed using a HR 20 Discovery Hybrid Rheometer. A frequency sweep with a fixed strain and a strain sweep with a fixed frequency were carried out to establish the optimal parameters. Based on these

preliminary experiments, a strain of 0.25% and a frequency of 2 Hz were identified as suitable conditions for mucin-agarose hydrogels. The acquired experimental data were analyzed using the TRIOS software. Two key parameters, namely the storage modulus ($G'$) and the loss modulus ($G''$), were measured. $G'$ reflects the material's elasticity, providing insights into its ability to store and recover energy. $G''$ represents the material's viscosity, indicating its resistance to flow. To ensure the reliability of the results, each gel sample underwent 3 to 4 repeated measurements, and the obtained values were averaged. Results of $G'$ and $G''$ were plotted in Prism 10.1.2. Error bars show standard deviations.

### Plaque assay in the presence of zanamivir

The role of neuraminidase in IAV spread was studied by fluorescent plaque assay upon exposure to neuraminidase inhibitor zanamivir. Zanamivir was diluted to final concentrations of 0, 1, 10 or 100 nM in plaque assay overlay medium (FBS-free DMEM-GlutaMAX-I, 1.2% avicel, 0.2% BSA and 1 μg/ml TPCK-Trypsin) and added to MDCK cells after infection with WSN and WSN-M1$_{Udorn}$ for 1 h. At 36 hpi, cells were chemically fixed and immunofluorescence staining of NP was performed. $R_{INF}$ of plaques from different zanamivir concentrations were determined according to the above-mentioned protocol.

### Plaque assay in the presence of neutralizing anti-HA antibodies (MEDI8852)

The effect of anti-HA broadly neutralizing antibodies (MEDI8852) (Kallewaard et al, 2016) on morphology-dependent IAV cell-to-cell spread was tested by fluorescent plaque assay in MDCK cells at 36 hpi. Antibody dilutions of 0, 0.5, 1, 2.5 and 5 nM were prepared in FBS-free DMEM-GlutaMAX-I medium and mixed with the avicel overlay medium. Infection with WSN:PAmScarlet or WSN-M1$_{Udorn}$:PAmScarlet, chemical fixation and immunofluorescence staining of NP were carried out according to the above-mentioned protocol. $R_{INF}$ [μm] and the relative reduction of $R_{INF}$ in the presence of increasing MEDI8852 concentration, normalized to control wells containing IAV-infected cells without antibody were calculated and three-parameter nonlinear curve fits were generated in Prism 10.1.2. From these fits, the IC50 (50% reduction of $R_{INF}$) was obtained.

### Live cell imaging of IAV-infected Calu-3 cells

For live cell imaging of fluorescent IAV plaques, Calu-3 cells infected with WSN:PAmScarlet or WSN-M1$_{Udorn}$:PAmScarlet were overlaid with 1% (w/v) agarose overlay medium (DMEM 2×, 2% P/S, 50 mM HEPES, 0.6% BSA and 4 μg/μl TPCK-Trypsin) and imaged with the ×5 objective of the CellDiscoverer 7 microscope under constant conditions of 37 °C, 20% O$_2$ and 5% CO$_2$. Cell nuclei were stained by addition of Hoechst into the overlay (1:1000 dilution). Images were acquired every 3 h over a course of 66 h, starting at 52 hpi or 79 hpi.

### Tracking and motion analysis of IAV-infected Calu-3 cells

To analyze cell migration within IAV plaques, we performed tracking and motion analysis of Calu-3 cells. First, the global sample drift in the time-lapse fluorescence microscopy images was computed using a method based on cross-correlation (Laine et al, 2019; Pylvanainen et al, 2023) for the Hoechst channel. The determined drift was then corrected in both the Hoechst and the mScarlet channel by shifting the images. Second, the fluorescently labeled cells in the microscopy images were tracked using a probabilistic particle tracking method (Ritter et al, 2021) based on multi-sensor data fusion and Bayesian filtering which combines Kalman filtering and particle filtering. Separate sensor models and sequential multi-sensor data fusion are used to integrate multiple detection-based and prediction-based measurements while considering different uncertainties. The measurements are obtained using elliptical sampling (Godinez and Rohr, 2015), and information from both past and future time points is exploited using Bayesian smoothing. Cell detection was performed by the spot-enhancing filter (SEF) (Sage et al, 2005) consisting of a Laplace-of-Gaussian (LoG) filter followed by intensity thresholding.

To analyze the motion of uninfected cells proximal to the infection plaque, a binary mask was created based on the mScarlet channel and computed trajectories within the mask were selected. First, segmentation of the infected cells in the mScarlet channel was performed by adaptive thresholding using a threshold defined by the mean intensity of the filtered image plus a factor times the standard deviation. The same factor was used for all images of an image sequence. Then, a binary mask of the infection plaque was created from the segmentation result using 40 iterations of binary dilation followed by hole filling using binary dilations and computing the binary sum over the complete image sequence. The largest connected component of the resulting binary mask was further dilated 120 times to create a mask for infection proximity. The final binary mask for analyzing infection-proximal uninfected cells was obtained by binary image subtraction of the infection mask from the infection-proximal mask. Computed trajectories in the Hoechst channel with most positions inside this mask were used for further analysis (Fig. EV4M). Additionally, the center of infection of an image sequence was determined as the center-of-mass of the segmentation result of the last image in the mScarlet channel. For uninfected control cells, the image center was used instead.

The motion of infected and uninfected cells was quantified by computing different motion properties from the computed trajectories. To improve the accuracy, only trajectories with a minimum duration of 12 h (corresponding to 4 time steps) were considered. The velocity over time for each trajectory was calculated and then averaged for infection-proximal uninfected cells as well as for cells in the mScarlet channel and cells in the Hoechst channel of image sequences without infection plaques. The distance migrated of a trajectory was computed as the distance between the first and last position, and the distance migrated towards the center was calculated as the difference between the distance of the first position to the center of infection and the distance of the last position to the center. In addition, to characterize the movement of cells relative to the center of infection, we computed the cosine deviation as the cosine of the angle between the direction from the first position to the last position and the direction from the first position to the center of infection (Fig. EV4M). The values of the cosine deviation lie in the range −1 to 1 with positive values close to 1 indicating motion towards the center of infection, and negative values indicating motion away from the center. Since the cell density of infection plaques strongly increases over time, only the first 54 h (corresponding to 19 time steps) were used to compute the above-mentioned motion properties to improve the accuracy. The motion properties were computed for each trajectory of an image sequence and then averaged over the image sequence.

### Entry time-course assay for spherical and filamentous IAV

To determine the entry half-time of spherical and filamentous IAV, A549 cells were seeded into 24-well plates (Corning) at a seeding density of $5 \times 10^4$ cells/well and incubated for 24 h at 37 °C and 5% $CO_2$. Cells were infected on ice with spherical or filamentous fluorescent reporter virus at an MOI of 3 in FBS-free DMEM-F12-GlutaMAX to guarantee synchronized infection. After 1 h on ice, cells were washed three times with cold PBS. PBS was replaced by FBS-free DMEM-F12-GlutaMAX and cells were incubated at 37 °C and 5% $CO_2$. At 0, 5, 10, 20, and 60 min post infection (mpi), cells were treated with 50 mM $NH_4Cl$ to block endosomal acidification and subsequently incubated at 37 °C and 5% $CO_2$. At 12 hpi, cells were fixed with 4% PFA in PBS for 30 min at RT. Immunolabeling of IAV-NP and DAPI staining was performed according to the above-described protocol.

### IAV entry assay in the presence of inhibitors

A549 cells were seeded into 24-well plates at a seeding density of $8 \times 10^4$ cells/well and incubated for 24 h at 37 °C and 5% $CO_2$. Cells were incubated with different drug concentrations of Dynasore (10, 50, 100 μM) or EIPA (40, 60, 80 μM) or DMSO in infection medium consisting of FBS-free DMEM-F12-GlutaMAX supplemented with 50 mM HEPES, 0.2% BSA for 1 h at 37 °C and 5% $CO_2$. Synchronized infection with spherical or filamentous WSN was performed on ice for 1 h at MOIs of 3 or 10, diluted in infection medium with inhibitor or DMSO. After virus attachment, virus inoculum was removed, and cells were washed 2 times with the corresponding cold inhibitor mix in infection medium. Plates were incubated with inhibitors at 37 °C and 5% $CO_2$ for 6 h, followed by washing with PBS and fixation with 4% PFA for 15 min at RT. Immunolabelling of NP and DAPI staining was performed according to the above-described protocol. Fluorescence microscopy images were acquired using the ×20 objective of the CellDiscoverer 7 microscope (pixel size $0.3450 \times 0.3450$ μm², 16 bit depth). Quantification of IAV infection for entry assays was performed in ImageJ/Fiji, using the StarDist plugin (Schmidt et al, 2018) to define regions of interest (ROIs). An intensity threshold of NP signal within these ROIs was set and cells with an average intensity above the threshold were counted as infected. Experiments were performed in three independent replicates.

### IAV entry assay in the presence of neutralizing antibodies

For the comparison of spherical and filamentous IAV cell entry efficiency in the presence of neutralizing anti-HA antibodies (MEDI8852), A549 cells were seeded into 24-well plates at a seeding density of $5 \times 10^4$ cells/well and incubated for 24 h at 37 °C and 5% $CO_2$. WSN and WSN-M1$_{Udorn}$ viruses were thawed on ice, diluted in FBS-free DMEM-F12-GlutaMAX and incubated with 0, 1, 5 or 10 nM MEDI8852 antibody at RT for 10 min. Cells were washed once with FBS-free DMEM-F12-GlutaMAX and infected with antibody-treated or untreated virus at an MOI of 3. After incubation at 37 °C and 5% $CO_2$ for 1 h, cells were washed 2 times with infection medium (FBS-free DMEM-F12-GlutaMAX supplemented with 50 mM HEPES, 0.2% BSA) and again incubated at 37 °C. Fixation was performed at 4 hpi with 4% PFA at RT for 15 min. Immunostaining of NP, fluorescence microscopy and quantification was performed as described in the section above.

### Serial passaging of IAV under neutralizing antibody pressure

The stability of spherical and filamentous IAV morphology exposed to 1 nM MEDI8852 antibody pressure was determined throughout five passages in cell culture. MDCK cells were seeded into 24-well plates at a density of $14 \times 10^4$ cells/well and incubated for 24 h at 37 °C and 5% $CO_2$ to reach ~80% confluency. Cells were washed once with FBS-free DMEM-GlutaMAX-I, infected with WSN or WSN-M1$_{Udorn}$ at MOI of 0.01 and incubated for 1 h at 37 °C and 5% $CO_2$. After washing twice with PBS, infection medium (FBS-free DMEM-GlutaMAX-I supplemented with 0.2% BSA and 1 μg/ml TPCK-Trypsin) containing 0 or 1 nM MEDI8852 antibody were added to the wells and plates were placed at 37 °C and 5% $CO_2$. Supernatants were harvested 48 hpi and cell debris were pelleted by centrifugation for 10 min at $1000 \times g$ and 4 °C. Virus-containing supernatants were snap-frozen in $LN_2$ and stored at −80 °C. After the first passage, viral titers were determined by plaque assay. This passaging process was performed five times. Supernatants of passages 1 and 5 were plunge-frozen and virion morphology was analyzed by cryo-EM as described in the section on morphology characterization.

### Sample preparation for cryo-electron microscopy

Glow-discharged EM grids (Quantifoil R1.2/20, holey $SiO_2$, Au 200 mesh) were placed into polydimethylsiloxane Sylgard 184 coated 35 mm dish (Corning) (Klein et al, 2021) and disinfected with 70% ethanol for 30 min. The dish was washed three times with DMEM-F12-GlutaMAX 10% FBS and 1% P/S. A549 cells were seeded at a density of $1.8 \times 10^5$ cells per dish and incubated at 37 °C and 5% $CO_2$ for 24 h. Virus was thawed on ice for 1 h and a dilution of $5 \times 10^6$ PFU/ml was prepared in FBS-free DMEM-F12-GlutaMAX. In all, 20 μl of virus dilution were pipetted onto parafilm placed in a 10 cm dish on a cooling plate at 4 °C. EM-grids were blotted on Whatman No.1 filter paper, placed onto the drop of virus and incubated for 30 min. Grids were washed 3 times with FBS-free DMEM-F12-GlutaMAX and placed in the incubator at 37 °C for 15–30 min. Cells were immediately plunge-frozen into liquid ethane using the following settings of an automatic EM GP2 plunge-freezing device: chamber temperature: 37 °C, humidity: 80%, back-side blotting: 3.5 s. Grids were stored in $LN_2$ until further processing.

### Cryo-focused ion beam milling (cryo-FIB milling)

Cryo-lamellae of infected cells were prepared by cryo-focused ion beam milling with an Aquilos 2 dual-beam cryo-focused ion beam-scanning electron microscope (cryo-FIB-SEM) with a cryo-stage at 180 °C. Grids were sputter coated before and after deposition of an organometallic platinum layer via the Gas Injection System (GIS). Using a gallium ion beam at an acceleration voltage of 30 keV, cells were semi-automatically milled at a stage angle of 15°. Three steps were milled using an adapted protocol for automated cryo-lamella preparation, including micro-expansion joints (Buckley et al, 2020; Wolff et al, 2019). The last two polishing-steps were performed manually to achieve a nominal lamella thickness of 150 nm.

### Cryo-electron tomography of IAV-infected cells

Cryo-ET data of lamellae from IAV-infected A549 cells were acquired with SerialEM (Mastronarde, 2005) using a Titan Krios cryo-TEM operated at 300 keV and equipped with a Quanta Imaging Filter with an energy filter slit set to 20 eV and a K3 direct electron detector. Grids were first mapped at ×8700 magnification (pixel spacing: 10.64 Å). Endosomes containing viruses were selected for tilt series acquisition at ×33,000 magnification (pixel

spacing: 2.67 Å) using PACEtomo (Eisenstein et al, 2023) in SerialEM following a dose-symmetric tilting scheme (Hagen et al, 2017) and the following settings: zero angle set to 8°, nominal tilt range from +68° to −52° and 3° increments, target defocus −3 μm, electron dose per record 3e⁻/Å². Drift correction of acquired movies was done with Motioncor2 (Zheng et al, 2017). Tomographic reconstruction was performed with AreTomo (Zheng et al, 2022). 3D segmentation was done using the Dragonfly software (Version 2022.2.0.12227).

## Data availability

Cryo-electron tomography data was deposited to Electron Microscopy Data Bank (EMDB): EMD-53448, EMD-53449. Additional data and material related to this publication may be obtained upon request.

The source data of this paper are collected in the following database record: biostudies:S-SCDT-10_1038-S44318-025-00481-6.

## Peer review information

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

## Acknowledgements

We thank Dr. Tijana Ivanovic for kindly providing MEDI8852 antibody and Dr. Michael D. Vahey for kindly providing a plasmid pCDNA3.1RG-WSN-M1$_{Udorn}$M2$_{WSN}$. We thank the Infectious Diseases Imaging Platform (IDIP) at the Center for Integrative Infectious Disease Research Heidelberg at Heidelberg University. We would like to acknowledge access to the infrastructure and support provided by the Cryo-EM Network at Heidelberg University (HDcryoNet), which is funded and supported by the German Research Foundation (DFG), the Federal Ministry of Science, Research and Arts of Baden-Württemberg, among others, within the framework of the Excellence Strategy of the Federal and State Governments of Germany. The authors would like to thank the Soft (bio)materials characterization Core Facility (Biomechanics) at IMSEAM and the Electron Microscopy Core Facility (EMCF) at Heidelberg University for their support. The authors gratefully acknowledge the data storage service SDS@hd supported by the Ministry of Science, Research, and the Arts Baden-Württemberg (MWK), the German Research Foundation (DFG) through grant INST 35/1314-1 FUGG and INST 35/1503-1 FUGG. This work was supported by a research grant from the Chica and Heinz Schaller Foundation (Schaller Research Group Leader Programme) and by the Deutsche Forschungsgemeinschaft (DFG, German Research Foundation): CF, EAC-A, CS-U, KR, and PC project no. 240245660-SFB1129; PC: 437060729. CS and MV were supported through the Max Planck School Matter to Life, funded by the German Federal Ministry of Education and Research (BMBF), the Max Planck Society, and the Flagship Initiative "Engineering Molecular Systems." We also acknowledge funding by the DFG under Germany's Excellence Strategy 2082/1-390761711 (3D Matter Made to Order) and the Carl Zeiss Foundation, as well as the European Research Council through the Consolidator Grant PHOTOMECH (no. 101001797). MV was also supported by Daimler and Benz Foundation (no. 32-09/24). For the publication fee we acknowledge financial support by Heidelberg University.

## Author contributions

**Sarah Peterl**: Conceptualization; Formal analysis; Investigation; Visualization; Methodology; Writing—original draft; Writing—review and editing. **Carmen M Lahr**: Conceptualization; Formal analysis; Investigation; Visualization; Methodology; Writing—review and editing. **Carl N Schneider**: Conceptualization; Formal analysis; Investigation; Visualization; Methodology; Writing—review and editing. **Janis Meyer**: Conceptualization; Formal analysis; Investigation; Visualization; Methodology; Writing—review and editing. **Xenia Podlipensky**: Formal analysis; Investigation; Methodology; Writing—review and editing. **Vera Lechner**: Formal analysis; Investigation; Methodology; Writing—review and editing. **Maria Villiou**: Conceptualization; Formal analysis; Investigation; Methodology; Writing—review and editing. **Larissa Eis**: Methodology; Writing—review and editing. **Steffen Klein**: Investigation; Methodology; Writing—review and editing. **Charlotta Funaya**: Supervision; Funding acquisition; Methodology; Writing—review and editing. **Elisabetta Ada Cavalcanti-Adam**: Resources; Supervision; Funding acquisition; Writing—review and editing. **Frederik Graw**: Formal analysis; Supervision; Writing—review and editing. **Christine Selhuber-Unkel**: Supervision; Funding acquisition; Validation; Writing—review and editing. **Karl Rohr**: Conceptualization; Supervision; Funding acquisition; Validation; Writing—review and editing. **Petr Chlanda**: Conceptualization; Supervision; Funding acquisition; Validation; Methodology; Writing—original draft; Project administration; Writing—review and editing.

Source data underlying figure panels in this paper may have individual authorship assigned. Where available, figure panel/source data authorship is listed in the following database record: biostudies:S-SCDT-10_1038-S44318-025-00481-6.

## Funding

## Disclosure and competing interests statement

The authors declare no competing interests.

# Expanded View Figures

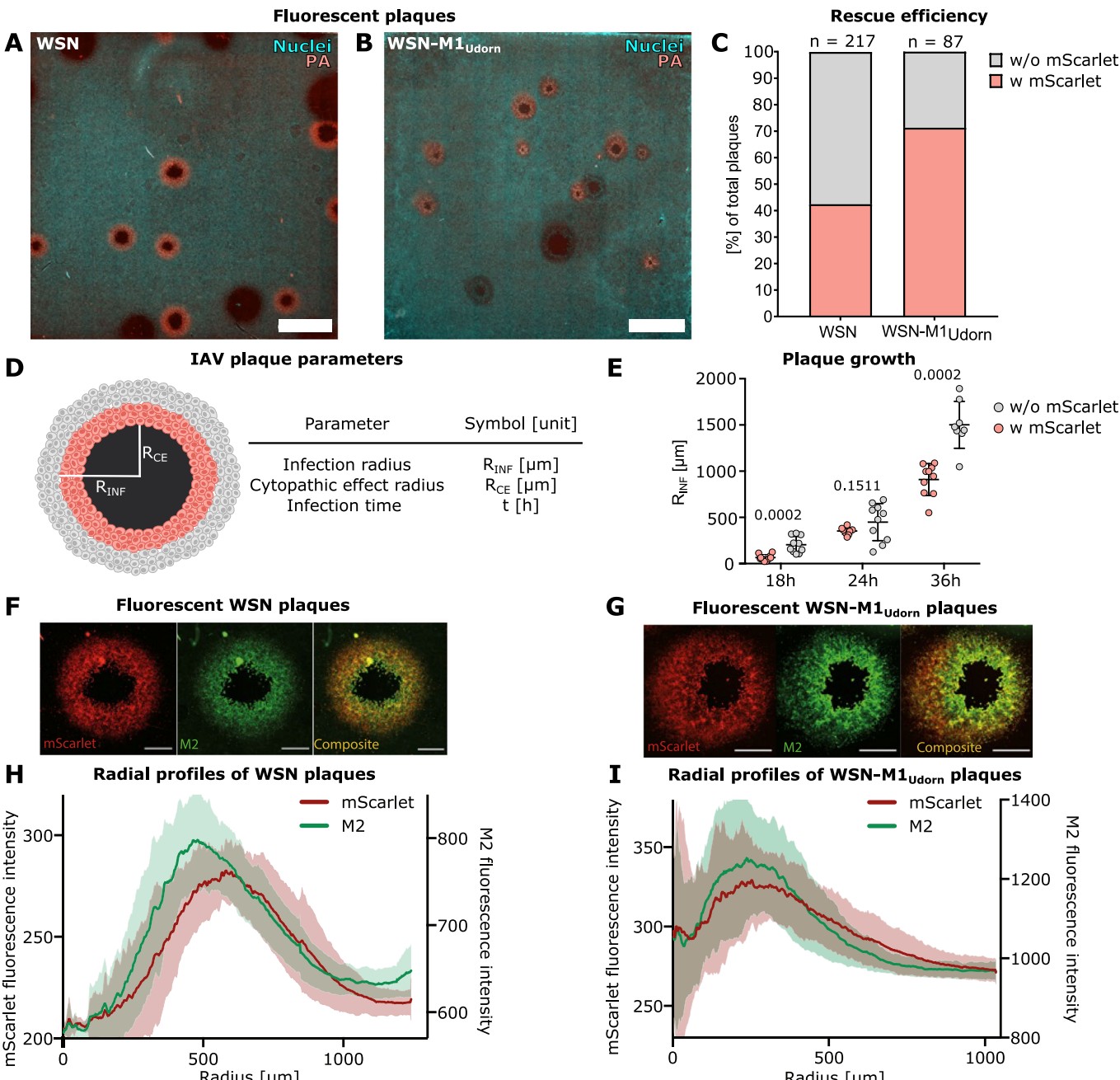

**Figure EV1. Characterization of fluorescent IAV plaques.**

(A) Exemplary images of plaques from MDCK cells infected with WSN:PAmScarlet at 36 hpi with PAmScarlet (red) and nucleus staining (cyan), scale bars: 3 mm. (B) Plaques from MDCK cells infected with WSN-M1$_{Udorn}$:PAmScarlet at 36 hpi with PAmScarlet (red) and nucleus staining (cyan), scale bars: 3 mm. (C) Percentages of plaques with (w) (red) and without (w/o) (gray) mScarlet signal for WSN:PAmScarlet ($n = 217$) and WSN-M1Udorn:PAmScarlet ($n = 87$), quantified at different time points. (D) Schematic representation of the zones within a fluorescent plaque and parameters for quantification of IAV spread. (E) Infection radius ($R_{INF}$) of plaques with (red) and without (gray) mScarlet quantified for 10 plaques per condition, at 18, 24, 36 hpi. Means and standard deviations are indicated. Exact *p* values for unpaired *t* tests: 2.151e-4, 1.511e-1, 2.324e-5. (F) Zoom-in of a fluorescent WSN:PAmScarlet plaque in MDCK cells from (A) with PAmScarlet (red), immunostained M2 (green), and a composite of both signals (yellow). The panel was intentionally duplicated from Fig. 1G to facilitate comparison between PAmScarlet and M2 stainings, scale bar: 500 μm. (G) Exemplary images of a fluorescent WSN-M1$_{Udorn}$:PAmScarlet plaque in MDCK cells with PAmScarlet (red), immunostained M2 (green), and a composite of both signals (yellow), scale bar: 500 μm. (H) Mean fluorescence intensities of PAmScarlet (red) and M2 (green) from radial profiles of 10 WSN:PAmScarlet plaques in MDCK cells. Standard deviations are indicated. (I) Mean fluorescence intensities of PAmScarlet (red) and M2 (green) from radial profiles of 10 WSN-M1$_{Udorn}$:PAmScarlet plaques in MDCK cells. Standard deviations are indicated.

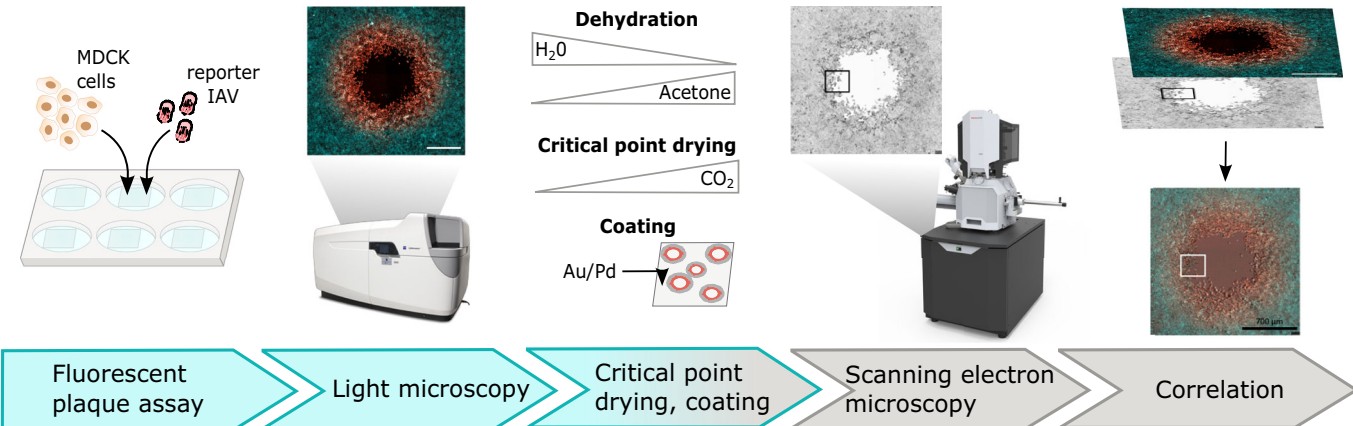

**Figure EV2.   Workflow of correlative light and scanning electron microscopy (CLSEM) for the study of plaque growth and IAV morphology.**

MDCK cells were seeded onto ITO-coated coverslips and infected with spherical or filamentous reporter IAV expressing PAmScarlet. Plaques were imaged by fluorescence microscopy. The plaque from Fig. 2A was reused here to demonstrate the CLSEM workflow. Samples were prepared for scanning electron microscopy (SEM) by dehydration with increasing acetone concentrations. Critical point drying and sputter coating with Au/Pd was performed. SEM overview maps of plaques were acquired for correlation with fluorescent images. High-magnification SEM images were acquired.

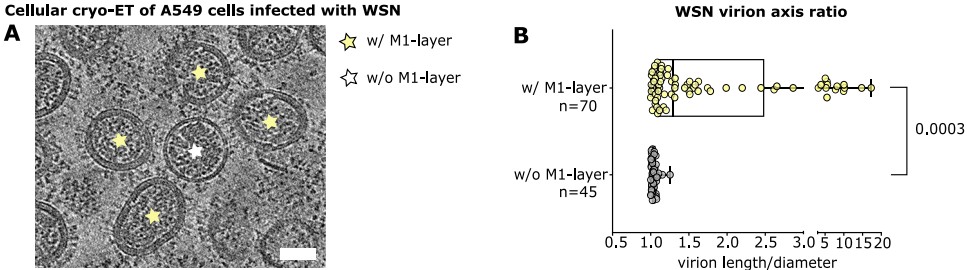

**Figure EV3.  Cellular cryo-ET of IAV-infected A549 cells.**

(A) Slice through a cryo-electron tomogram showing spherical WSN particles in endosomal compartments of an infected A549 cell at 15–30 min post infection.Virions with assembled matrix protein 1 layer (w/ M1-layer) are highlighted with yellow stars. One virion without (w/o) M1-layer is highlighted with a white star, scale bar: 50 nm. (B) WSN virion length/diameter ratio of particles w/ M1-layer (yellow) and w/o M1-layer (gray) within endosomes of infected A549 cells, quantified from cryo-electron tomograms. Boxes show quartiles with median lines and min to max values. Significance analysis was done by Mann–Whitney $U$-test. Exact $p$ value: 3.2e-4.

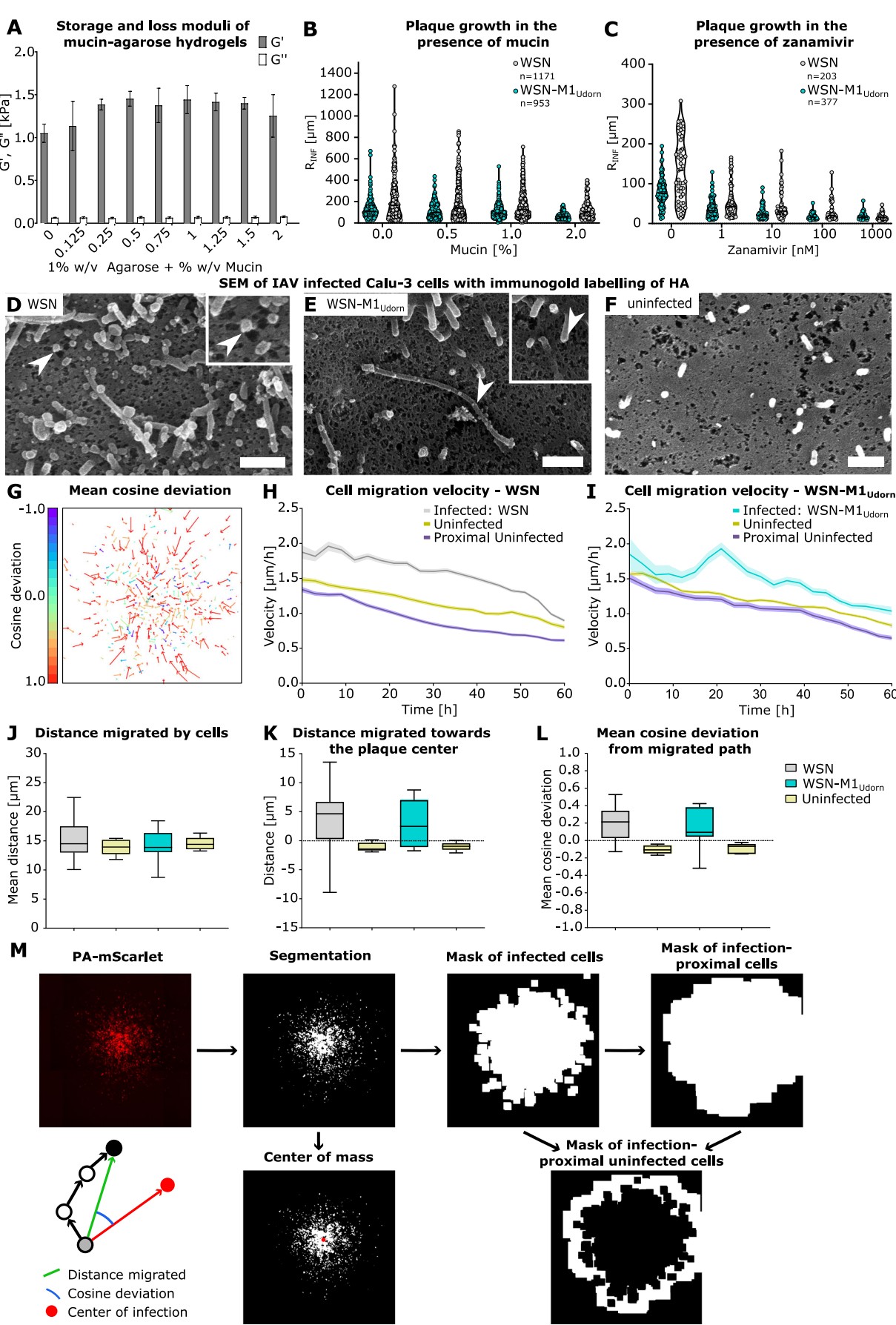

**Figure EV4.** **IAV plaque growth in the presence of mucin or zanamivir, immunoelectron microscopy of IAV and Calu-3 cell migration.**

(A) Rheological properties of agarose hydrogels in the presence of increasing mucin concentrations. The storage modulus ($G'$, gray) reflects elasticity and the loss modulus ($G''$, white) represents viscosity. Bars show averages of 3–4 repeated measurements in 5 different mucin-agarose hydrogels with indicated standard deviations. (B) $R_{INF}$ of WSN (gray) and WSN-M1$_{Udorn}$ (cyan) in the presence of increasing mucin concentrations at 36 hpi in MDCK cells. (C) $R_{INF}$ of WSN (gray) and WSN-M1$_{Udorn}$ (cyan) in the presence of increasing zanamivir concentrations at 36 hpi in MDCK cells. (D) Scanning electron microscopy (SEM) of Calu-3 cells infected with WSN:PAmScarlet, scale bar: 500 nm. (E) SEM of Calu-3 cells infected with WSN-M1$_{Udorn}$:PAmScarlet, fixed at 4 days post infection. White arrowheads indicate 20 nm immunogold labeling of hemagglutinin (HA), scale bar: 500 nm. (F) SEM of uninfected Calu-3 cells, scale bar: 500 nm. (G) Migrated paths of Calu-3 cells tracked from their first to last position of time-lapse movies. Color code of arrows represent the cosine deviation from migrated path, where +1 indicates migration to the plaque center and −1 indicates migration away from the center. (H) Mean velocity of cell migration trajectories for WSN:PAmScarlet infected and uninfected Calu-3 cells. Standard errors are indicated. The time point 0 h corresponds to 52 hpi. (I) Mean velocity of cell migration trajectories for WSN-M1$_{Udorn}$:PAmScarlet infected and uninfected Calu-3 cells. Standard errors are indicated. The time point 0 h corresponds to 52 hpi. (J) Mean migrated distance of Calu-3 cells from their first to last position. Boxes show quartiles with median lines and min to max values. (K) Migrated distance of Calu-3 cells towards the focus center. For each movie, the distances are averaged over all trajectories. Boxes show quartiles with median lines and min to max values. (L) Mean cosine deviation of cell movement from migrated path with positive values indicating migration towards the center of infection. Boxes show quartiles with median lines and min to max values. Color code of plots H-M: gray: infected with WSN:PAmScarlet, cyan: infected with WSN-M1$_{Udorn}$:PAmScarlet, yellow: uninfected cells, blue: proximal uninfected cells. (M) Workflow of tracking and motion analysis for cells within IAV foci, using foci from Fig. 4G as an example. IAV-infected cells expressing PAmScarlet (red) were segmented by adaptive thresholding. From the last segmented image of the time-lapse series, the center of infection was determined as the center of mass. Based on the segmentation, binary masks of infection foci were created by binary dilation and hole filling. A mask for infection-proximal cells was created by further dilating this mask 120 times. To obtain a mask for uninfected cells in proximity to IAV foci the mask of infected cells was subtracted from the mask of infection-proximal cells. Cell migration was determined by probabilistic particle tracking. The distance migrated from the first to the last position (green) and the cosine of the angle (blue) between the directions from the first to the last position (green) and the first position and the plaque center (red) were computed.

**Figure EV5.  Effect of neutralizing MEDI8852 antibodies on IAV plaque size.**

Reduction of infection radius ($R_{INF}$) of WSN-M1$_{Udorn}$ plaques relative to WSN plaques in MDCK cells in the presence of indicated MEDI8852 concentrations. Relative reduction [%] = 100 − (Radius of WSN-M1$_{Udorn}$/Radius of WSN × 100). Mean and standard deviations are indicated for three independent experiments. *P* values were calculated using one-sided Student's *t* test comparing each concentration with the untreated control. Exact *p* values: 1.670e-1, 3.471e-1, 3.930e-2, 4.803e-6.

