## [Peer Review File · The EMBO Journal]

Morphology-dependent entry kinetics and spread of influenza A virus

Sarah Peterl, Carmen Lahr, Carl Schneider, Janis Meyer, Xenia Podlipensky, Vera Lechner, Maria Villiou, Larissa Eis, Steffen Klein, Charlotta Funaya, Elisabetta Cavalcanti-Adam, Frederik Graw, Christine Selhuber-Unkel, Karl Rohr, and Petr Chlanda

Corresponding author: Petr Chlanda (petr.chlanda@bioquant.uni-heidelberg.de)

Review Timeline:

Submission Date:	8th Mar 25
Editorial Decision:	11th Apr 25
Revision Received:	25th Apr 25
Accepted:	14th May 25

Editor: Ieva Gailite

Transaction Report:

Review
COMMONS

This manuscript was transferred to The EMBO Journal following peer review at Review Commons.

Review #1**1. Evidence, reproducibility and clarity:****Evidence, reproducibility and clarity (Required)**

This manuscript by Peterl and colleagues seeks to understand the long-standing observation that influenza A virus generally exhibits a filamentous phenotype in vivo which is lost upon serial passaging in vitro or in embryonated chicken eggs. In addressing this question, the authors perform a detailed quantitative comparison of how filamentous and spherical strains of influenza spread in cell culture in the presence or absence of perturbations including neutralizing antibodies, mucin, and disruption of cell-cell junctions.

The manuscript reports several observations that will be of interest to researchers in the area of influenza virus morphology and spread. Using a combination of imaging modalities, the authors convincingly demonstrate that spherical strains of influenza virus produce larger plaques than filamentous strains that are isogenic except for mutations in M1. The authors show that this is at least partly attributable to differences in entry kinetics. The authors also recapitulate a prior finding that filamentous viruses are more resistant to neutralizing antibodies than spherical ones. In most cases, the authors' claims are supported by the data presented. A few partial exceptions are noted below.

The paper would be strengthened by a clearer description of some of the experimental approaches which lack important details in some instances. The scope of the paper is also limited somewhat by the use of immortalized cell lines that lack physiological features of the airway epithelium. Although this limitation is understandable from a technical standpoint, a discussion of these limitations should be included. Specific comments are listed below.

****Major Points****

In Figure 4, it is not stated at what time the cell density is measured in panel B, and how this might change across the time points sampled in panel C. This would make the experiment difficult to reproduce. This could be a very important consideration if the cells reach confluency soon after the infection is initiated, since the plaque sizes seem statistically similar out to 24hpi in 4B.

In Figure 4F, it appears that plaque sizes for M1Ud are less affected by mucin than M1WSN

plaques at all concentrations tested. However, the authors conclude that "mucin did not show any IAV morphology-dependent inhibitory effect as indicated by the slopes of linear fits of the plaque diameters" (Line 265). I understand that the authors are looking for dose-dependent effects, but it is not clear to me why an analysis based on the slope is preferable, especially when the response to mucins may not be linear. How does the availability of IAV receptors in the porcine gastric mucin used here compare to human airway mucins? Finally, the authors should clarify the number of replicates for this experiment.

One key difference between the cells used here and the airway epithelium is the presence of multiciliated cells that could alter viral transport in ways that depend on morphology and may be difficult to predict. I appreciate that this concept is outside the scope of the current work, but it is an important point that warrants mention.

****Minor Points****

It is somewhat unclear what is being captured in the data in Figure 5D-I. I assume that the cell surfaces that are imaged here are from infected cells within the plaque. If this is the case, it is difficult to tell whether the particles that are being quantified are incoming viruses or viruses that are currently budding. MEDI8852 is a stalk-binding antibody which would not be expected to inhibit viral attachment. This is unlikely to change the interpretation since the data shows differences between spherical and filamentous strains. However, a clearer description of this data would be helpful.

For experiments in Calu-3 cells, is trypsin added to the culture media following infection? If not, what percentage of HA is proteolytically cleaved? I would expect these cells to express activating proteases, but if activation is less efficient, this could favor the filamentous strain (as discussed in ref 49).

The schematic in Figure 4D illustrates mucins as tethered to the cell surface. This does not reflect the experiments in Figure 4E and F, where secreted mucins are added to the overlay media.

There are a few small typos. Line 61: "to results in" and Line 111: "neutralizing antibodies against hemagglutinin are more effectively blocking virions with spherical morphology."

2. Significance:

Significance (Required)

A strength of this manuscript is the quantitative rigor of the approaches used, which reveal interesting differences in the spread of filamentous and spherical influenza. These differences are compelling, but are limited somewhat in their significance by the difficulty of evaluating whether or not some of the observations would be preserved in differentiated airway epithelial cells. The authors do not over-generalize their conclusions, but more detailed discussion of these potential limitations is warranted.

3. How much time do you estimate the authors will need to complete the suggested revisions:

Estimated time to Complete Revisions (Required)

(Decision Recommendation)

Less than 1 month

4. Review Commons values the work of reviewers and encourages them to get credit for their work. Select 'Yes' below to register your reviewing activity at Web of Science Reviewer Recognition Service (formerly Publons); note that the content of your review will not be visible on Web of Science.

Yes

Review #2

1. Evidence, reproducibility and clarity:

Evidence, reproducibility and clarity (Required)

****Summary:****

This manuscript by Peteryl and colleagues explores the question of why some influenza viruses (typically those that have been recently isolated from animals, though also the Udorn strain) produce filamentous particles, while influenza viruses that have been adapted to eggs or cell culture form spherical particles. This is a long standing question in the influenza field, and the authors have used a nice set of new tools and approaches to shed light on this question. They created mScarlet labelled viruses that produce spherical (WSN) or predominantly filamentous (WSN with an M segment from Udorn) virions, but share the same glycoproteins. While this approach is not novel (the fact that the segment 7 of Udorn drives a filamentous phenotype has been previously demonstrated), the authors

used these viruses in an elegant series of experiments to look at the rate of cell to cell spread within a plaque to show that the spherical viruses spread more quickly. The authors then explored the effect of cell density, inhibitors designed to inhibit different routes of viral entry, and the presence of neutralizing antibody. The experiments are thoughtfully designed, and the electron microscopy in particular is beautifully done. In general, the conclusions are supported by the data, though the specific claim that filamentous viruses have an advantage in viral entry in the presence of neutralizing antibody would be significantly strengthened by performing the specific entry assay the authors employ earlier in the manuscript.

****Major comments:****

The key conclusions are largely convincing, though the authors should perform the entry assays they employ in figure 3 (measuring the kinetics of entry and the efficiency of entry) to determine whether the delay in cell to cell spread they observe for spherical viruses in the presence of neutralizing antibody is due specifically to the effect on entry. I also am concerned about the method used to determine that the antibody treatment in Fig 5D-H results in a difference in the number of virions produced. While I appreciate that SEM is time consuming and difficult to quantify, counting the number of virions seen in a single field of view from 7 or 12 cells does not provide a robust foundation to support the central claim of the paper, that the difference in speed of filamentous and spherical viral spread is due to a difference in their ability to support viral entry in the presence of neutralizing antibody . If the authors wish to count virions produced by the WSN/WSN M-Udorn viruses in the presence/absence of neutralizing antibody it would be sensible to perform a synchronized high MOI infection and measure infectious titer by plaque assay (as this would be able to quickly and easily measure millions of virions produced by hundreds of thousands of cells).

The two entry assays could be done in parallel and I anticipate them to take ~3 days per replicate (a day to seed, a day to infect/add NH₄Cl at the indicated time points and fix, a day to image and analyze data). Similarly, infected cells at high MOI in the presence/absence of nAb, collecting viral supernatants, and titering by plaque assay should take ~one week. The reagents to perform these experiments are already in hand, and as the costs will be limited to standard tissue culture reagents, using a microscopy set up the authors already possess. The experiments throughout the paper are well described, with appropriate methodological detail and statistical analysis.

****Minor comments:****

- Viruses without the mScarlet spread faster, the WSN-Udorn has more viruses with mScarlet than the WSN does so how do we know that some of the difference isn't down to that?
- While Calu3 cells are reported to make mucus the authors should verify the expression of relevant mucus proteins in their hands, and this phenotype can be variable depending on culture conditions.
- In 5F and I does 'mock' mean no antibody or no virus?
- The authors should either include data to support the claim in line 410: "Our data provide further evidence that IAV filamentous morphology is lost to accelerate cell-to-cell spread by faster entry kinetics and to achieve higher entry efficiency" or reword this sentence, since at present this manuscript does not include experiments demonstrating the loss of filamentous morphology in tissue culture of the WSN-M1 Udorn virus.

2. Significance:

Significance (Required)

The data and conclusions presented in this manuscript are exciting and novel, and should be of high interest to virologists and cell biologists. The work builds on (and appropriately references) prior work in the field of influenza particle shape by the Lamb, Barclay, Garcia-Sastre, Vahey, Fletcher and Ivanovic groups. It provides new information and techniques to show that spherical virions spread faster than filamentous virions within plaques, and this advantage is not negated by cell density, the presence of mucus, or different entry inhibitors but is significantly reduced in the presence of neutralizing antibodies. It also includes other useful observations to the field (the fact that infected Calu3 cells migrate to the center of infected plaques, the fact that the entry kinetics and success rate of filaments is lower compared to spheres).

Expertise: virology, influenza, virion morphology, cell biology

3. How much time do you estimate the authors will need to complete the suggested revisions:

Estimated time to Complete Revisions (Required)

(Decision Recommendation)

Less than 1 month

4. Review Commons values the work of reviewers and encourages them to get credit for their work. Select 'Yes' below to register your reviewing activity at Web of Science

Reviewer Recognition Service (formerly Publons); note that the content of your review will not be visible on Web of Science.

Yes

Review #3

1. Evidence, reproducibility and clarity:

Evidence, reproducibility and clarity (Required)

The manuscript by Peterl et al. deals with the still interesting question of why influenza A viruses are filamentous in natural isolates but adopt a spherical phenotype in cell culture. The authors generated recombinant IAV reporter viruses that display identical antigenic (HA and NA) surfaces but differ in their morphology due to expression of an M1 protein that confers a spherical or filamentous phenotype. The data show that spherical viruses exhibit increased entry kinetics and spread faster in cell culture compared to filamentous viruses and that this is also the case in the presence of mucins and at a low cell density.

Interestingly, the authors found that spherical viruses are more efficiently blocked by neutralizing HA antibodies than filamentous viruses, providing an interesting advantage for the filamentous phenotype of natural IAV isolates due to antibody pressure.

The manuscript is of the usual excellent quality of the working group of Petr Chlanda and the data are very interesting. The experiments are well thought out and the results are comprehensible, convincing and visually very clear. The fact that a current preprint also describes that neutralizing antibodies drives filamentous virus formation (as mentioned by the authors in the discussion) does not diminish the message and quality of this work.

There were a few minor open questions that came to mind that could be included in the discussion:

The authors found that the filamentous morphology was stable throughout multiple rounds of infection during plaque formation. Is this still the case even with multiple passages (e.g 10x) in cell culture or does the number of spherical particles increase at some point?

The filamentous virus spreads slower in cell culture. Does NA play a role here? NA is probably distributed differently on the surface of filamentous viruses (at the tips) than on spherical viruses?

2. Significance:

Significance (Required)

The manuscript is of the usual excellent quality of the working group of Petr Chlanda and the data are very interesting. The experiments are well thought out and the results are comprehensible, convincing and visually very clear.

3. How much time do you estimate the authors will need to complete the suggested revisions:

Estimated time to Complete Revisions (Required)

(Decision Recommendation)

Less than 1 month

No

Full Revision

Manuscript number: RC-2024-02635

Corresponding author(s): Petr, Chlanda

1. General Statements [optional]

We sincerely thank all reviewers for taking the time to review our manuscript and for providing insightful comments and suggestions. Your feedback has been invaluable in improving the quality and clarity of our work. We have included the following new experiments:

- a. Effect of neutralizing antibody on IAV entry
- b. Passaging experiment in the presence of neutralizing antibody and morphology assessment by cryo-EM
- c. Effect of neuraminidase activity on IAV morphology-dependent spread
- d. Mucin expression in Calu-3 cells

Reviewer #1

Evidence, reproducibility and clarity

This manuscript by Peterl and colleagues seeks to understand the long-standing observation that influenza A virus generally exhibits a filamentous phenotype in vivo which is lost upon serial passaging in vitro or in embryonated chicken eggs. In addressing this question, the authors perform a detailed quantitative comparison of how filamentous and spherical strains of influenza spread in cell culture in the presence or absence of perturbations including neutralizing antibodies, mucin, and disruption of cell-cell junctions.

The manuscript reports several observations that will be of interest to researchers in the area of influenza virus morphology and spread. Using a combination of imaging modalities, the authors convincingly demonstrate that spherical strains of influenza virus produce larger plaques than filamentous strains that are isogenic except for mutations in M1. The authors show that this is at least partly attributable to differences in entry kinetics. The authors also recapitulate a prior finding that filamentous viruses are more resistant to neutralizing antibodies than spherical ones. In most cases, the authors' claims are supported by the data presented. A few partial exceptions are noted below.

The paper would be strengthened by a clearer description of some of the experimental approaches which lack important details in some instances. The scope of the paper is also limited somewhat by the use of immortalized cell lines that lack physiological features of the airway epithelium. Although this limitation is understandable from a technical standpoint, a discussion of these limitations should be included. Specific comments are listed below.

Major Points

In Figure 4, it is not stated at what time the cell density is measured in panel B, and how this might change across the time points sampled in panel C. This would make the experiment difficult to reproduce. This could be a very important consideration if the cells reach confluency soon after the infection is initiated, since the plaque sizes seem statistically similar out to 24hpi in 4B.

Thank you for your comment on cell densities in Figure 4 B. We agree that the quantification of cell confluency across the time points is crucial in this context. Furthermore, we recognize that counting the number of nuclei within a well is not the most accurate method for comparing the two cell lines. We now provide measurements of relative cell density based on plasma membrane labelling for uninfected MDCK-WT and MDCK- α -Catenin-KO cells at 24h and 48h for three biological replicates (Figure 4 A and B). These data show that MDCK- α -Catenin-KO have lower confluency (area=229.69 μm^2) at 48 h compared to MDCK-WT cells (area=361.24 μm^2). While the confluency of MDCK-WT cells was > 95% at both time points, MDCK- α -Catenin-KO cells did not reach 70% confluency, which reflects the lack of adherens junctions in these cells.

In Figure 4F, it appears that plaque sizes for M1Ud are less affected by mucin than M1WSN plaques at all concentrations tested. However, the authors conclude that "mucin did not show any IAV morphology-dependent inhibitory effect as indicated by the slopes of linear fits of the plaque diameters" (Line 265). I understand that the authors are looking for dose-dependent effects, but it is not clear to me why an analysis based on the slope is preferable, especially when the response to mucins may not be linear. How does the availability of IAV receptors in the porcine gastric mucin used here compare to human airway mucins? Finally, the authors should clarify the number of replicates for this experiment.

Thank you for pointing out that the data representation of IAV WSN and WSN-M1_{Udorn} plaque growth in the presence of mucin (Figure 4 C) lacked clarity. We agree and removed the regression fitting and, instead, show all individual plaque sizes (Extended Figure 4 B). We now provide relative reduction of plaque sizes compared between WSN and WSN-M1_{Udorn} plaques at each mucin concentration using 3 or 4 independent experiments (Figure 4 E). This did not reveal that there was a significant reduction in plaque size change between WSN and WSN-M1_{Udorn} in the absence or presence of mucin. We changed our conclusion: "mucin did not show an IAV morphology-dependent inhibitory effect as indicated by the relative plaque size decrease of WSN-M1_{Udorn} compared to WSN across the mucin concentrations" (Line 278).

We have included information on the mucus composition and receptor availability in the discussion: "Notably, we used porcine gastric mucin, which might differ structurally and in the sialic acid linkage types compared to human mucins (Nordman et al., 2002, doi: 10.1042/bj3640191; Zhang et al., 2021, doi: 10.1007/s10719-021-10014-y). However, both in the porcine stomach and human airway, MUC5AC molecules are the predominant gel-forming mucins." (Graigner et al., 2006, 10.1007/s11095-006-0255-0) (Line 436).

One key difference between the cells used here and the airway epithelium is the presence of multiciliated cells that could alter viral transport in ways that depend on morphology and may be

difficult to predict. I appreciate that this concept is outside the scope of the current work, but it is an important point that warrants mention.

We have now included fluorescence microscopy data using anti-MUC5AC antibody to assess mucin production in Calu-3 cells. Importantly, we could demonstrate that Calu-3 cells used in our study express mucins (Figure 4 D). We acknowledge that the absence of multiciliated cells is a limitation and plan to address this in future studies by using air-liquid interface cultures and by incorporating primary human bronchial cells. We established a transwell Calu-3 cell culture under air-liquid interface (ALI) conditions, which allowed for cell polarization. The apical surface of Calu-3 cells grown in an ALI culture contains more mucin than in liquid-covered unpolarized cultures. We plan to adapt and further develop a correlative imaging workflow to be able to assess spread in transwells in a separate study, as this is technically more challenging. We have included this in the discussion (Line 440-444).

Minor Points

It is somewhat unclear what is being captured in the data in Figure 5D-I. I assume that the cell surfaces that are imaged here are from infected cells within the plaque. If this is the case, it is difficult to tell whether the particles that are being quantified are incoming viruses or viruses that are currently budding. MEDI8852 is a stalk-binding antibody which would not be expected to inhibit viral attachment. This is unlikely to change the interpretation since the data shows differences between spherical and filamentous strains. However, a clearer description of this data would be helpful.

We appreciate your constructive feedback. Figure 5 captures the effect of HA-stalk-binding MEDI8852 antibodies on IAV spread and morphology. While this antibody does not prevent receptor binding, it blocks membrane fusion and exerts pressure on the viruses, which, based on our hypothesis, can be overcome by increasing the number of HA on the surface of filamentous viruses. This is now also confirmed in Figure 5B showing that entry of spherical viruses is more sensitive to MEDI8852 than entry of filamentous viruses above concentration of 5 nM.

SEM images of IAV plaques in MDCK cells in the presence of 1 nM MEDI8852 antibody show that viral morphology is not altered by antibody pressure. We agree that this method provides information on IAV morphology but does not allow us to distinguish between incoming or budding viruses. However, virus entry is fast, and IAV release from plasma membrane is slow as obvious from transmission electron microscopy studies showing large quantities of budding virions connected to plasma membrane by budding neck (example: DOI: 10.1099/vir.0.036715-0). Hence, it can be assumed that the majority of viruses captured by SEM on the cell surface are budding viruses. We have included this in the discussion (Line 409-414).

Nevertheless, to further address this limitation, we now provide a more robust analysis of IAV particle numbers and morphologies from supernatants of serial passaging in MDCK cells under MEDI8852 antibody pressure, using cryo-EM (Fig. 5 D, E). In accordance with the SEM data,

we did not observe morphological changes of IAV in the presence of the antibody.

For experiments in Calu-3 cells, is trypsin added to the culture media following infection? If not, what percentage of HA is proteolytically cleaved? I would expect these cells to express activating proteases, but if activation is less efficient, this could favor the filamentous strain (as discussed in ref 49).

Thank you for this comment. Yes, trypsin was added to the medium of Calu-3 cells during infection. We included this in the methods section.

The schematic in Figure 4D illustrates mucins as tethered to the cell surface. This does not reflect the experiments in Figure 4E and F, where secreted mucins are added to the overlay media.

We agree, and we removed the schematic representation of mucins in Figure 4D, instead with show data on mucin production in Calu-3 cells (Figure 4 D).

There are a few small typos. Line 61: "to results in" and Line 111: "neutralizing antibodies against hemagglutinin are more effectively blocking virions with spherical morphology."

We corrected the typo in line 61 and changed the phrasing of lines 111-112 for more clarity.

Significance

A strength of this manuscript is the quantitative rigor of the approaches used, which reveal interesting differences in the spread of filamentous and spherical influenza. These differences are compelling, but are limited somewhat in their significance by the difficulty of evaluating whether or not some of the observations would be preserved in differentiated airway epithelial cells. The authors do not over-generalize their conclusions, but more detailed discussion of these potential limitations is warranted.

As mentioned above, we agree that a differentiated airway is important; however, assessing determining factors responsible for inhibition might be difficult due to the high complexity of the culture composed of different cells. The presented methods allow quantitatively assessing individual factors, which provides benefits. Hence, both approaches are valid and important.

Reviewer #2

Evidence, reproducibility and clarity

Summary:

This manuscript by Peteryl and colleagues explores the question of why some influenza viruses (typically those that have been recently isolated from animals, though also the Udorn strain) produce filamentous particles, while influenza viruses that have been adapted to eggs or cell culture form spherical particles. This is a long standing question in the influenza field, and the authors have used a nice set of new tools and approaches to shed light on this question. They created mScarlet labelled viruses that produce spherical (WSN) or predominantly filamentous (WSN with an M segment from Udorn) virions, but share the same glycoproteins. While this approach is not novel (the fact that the segment 7 of Udorn drives a filamentous phenotype has been previously demonstrated), the authors used these viruses in an elegant series of experiments to look at the rate of cell to cell spread within a plaque to show that the spherical viruses spread more quickly. The authors then explored the effect of cell density, inhibitors

designed to inhibit different routes of viral entry, and the presence of neutralizing antibody. The experiments are thoughtfully designed, and the electron microscopy in particular is beautifully done. In general, the conclusions are supported by the data, though the specific claim that filamentous viruses have an advantage in viral entry in the presence of neutralizing antibody would be significantly strengthened by performing the specific entry assay the authors employ earlier in the manuscript.

Major comments:

The key conclusions are largely convincing, though the authors should perform the entry assays they employ in figure 3 (measuring the kinetics of entry and the efficiency of entry) to determine whether the delay in cell to cell spread they observe for spherical viruses in the presence of neutralizing antibody is due specifically to the effect on entry. I also am concerned about the method used to determine that the antibody treatment in Fig 5D-H results in a difference in the number of virions produced. While I appreciate that SEM is time consuming and difficult to quantify, counting the number of virions seen in a single field of view from 7 or 12 cells does not provide a robust foundation to support the central claim of the paper, that the difference in speed of filamentous and spherical viral spread is due to a difference in their ability to support viral entry in the presence of neutralizing antibody. If the authors wish to count virions produced by the WSN/WSN M-Udorn viruses in the presence/absence of neutralizing antibody it would be sensible to perform a synchronized high MOI infection and measure infectious titer by plaque assay (as this would be able to quickly and easily measure millions of virions produced by hundreds of thousands of cells).

Thank you very much for the suggestion to perform an entry assay in the presence of a neutralizing antibody to determine whether the antibody acts at the level of viral entry. We now provide data on the entry efficiency of WSN and WSN-M1_{Udorn} in the presence of increasing MEDI8852 concentrations (Figure 5 B). The results show that entry of the WSN spherical viruses are more affected by MEDI8852 at 5 nM and 10 nM, compared to WSN-M1_{Udorn}, suggesting that the reduced plaque growth presented in Figure 5 C reflects an inhibition of IAV entry.

We agree that the quantification of virions at the surface of 7-12 cells in SEM images is not a robust method. Therefore, we removed the quantification as it is technically very time-consuming to obtain a large enough dataset or to perform statistical power analysis on how many cells would need to be screened. We additionally performed a serial passaging experiment of WSN and WSN-M1_{Udorn} under antibody pressure, providing a more robust analysis of IAV particle numbers and morphologies from supernatants using cryo-EM (Fig. 5 D, E). By quantifying the length/diameter ratio of at least 80 virions per condition, we observed that both IAV morphologies remained stable in the presence of the antibody after five passages.

The two entry assays could be done in parallel, and I anticipate them to take ~3 days per replicate (a day to seed, a day to infect/add NH₄Cl at the indicated time points and fix, a day to image and analyze data). Similarly, infected cells at high MOI in the presence/absence of nAb, collecting viral supernatants, and titering by plaque assay should take ~one week. The reagents to perform these experiments are already in hand, and as the costs will be limited to

standard tissue culture reagents, using a microscopy set up the authors already possess. The experiments throughout the paper are well described, with appropriate methodological detail and statistical analysis.

Minor comments:

- Viruses without the mScarlet spread faster, the WSN-Udorn has more viruses with mScarlet than the WSN does so how do we know that some of the difference isn't down to that?

Thank you for this important question. It is correct that viruses without mScarlet spread faster. We used WSN mScarlet viruses for CLSEM and live cell imaging of Calu-3 cells. To ensure that the observed differences in viral spread kinetics were not attributable to the presence or absence of mScarlet but to viral morphology, we conducted additional immunofluorescence staining for viral nucleoprotein (NP) or matrix protein 2 (M2) (Extended Figure 1 H-I). This allowed us to account for all viral plaques, including those that were not mScarlet-positive. This way we obtained data for our experiments with MDCK- α -Catenin-KO cells, mucin, zanamivir and MEDI8852 (Figure 4 and 5).

- While Calu3 cells are reported to make mucus the authors should verify the expression of relevant mucus proteins in their hands, and this phenotype can be variable depending on culture conditions.

Thank you for highlighting this important point. We verified the expression of MUC5AC in Calu-3 cells grown on cover slips and observed MUC5AC expression in distinct puncta (Figure 5 D).

- In 5F and I does 'mock' mean no antibody or no virus?

We apologize for the imprecise nomenclature in Figure 5 F and changed the Figure description.

- The authors should either include data to support the claim in line 410: "Our data provide further evidence that IAV filamentous morphology is lost to accelerate cell-to-cell spread by faster entry kinetics and to achieve higher entry efficiency" or reword this sentence, since at present this manuscript does not include experiments demonstrating the loss of filamentous morphology in tissue culture of the WSN-M1 Udorn virus.

Thank you, we agree and modified the sentence.

Significance

The data and conclusions presented in this manuscript are exciting and novel, and should be of high interest to virologists and cell biologists. The work builds on (and appropriately references) prior work in the field of influenza particle shape by the Lamb, Barclay, Garcia-Sastre, Vahey, Fletcher and Ivanovic groups. It provides new information and techniques to show that spherical virions spread faster than filamentous virions within plaques, and this advantage is not negated by cell density, the presence of mucus, or different entry inhibitors but is significantly reduced in the presence of neutralizing antibodies. It also includes other useful observations to the field (the fact that infected Calu3 cells migrate to the center of infected plaques, the fact that the entry kinetics and success rate of filaments is lower compared to spheres).
Expertise: virology, influenza, virion morphology, cell biology

Reviewer #3

Evidence, reproducibility and clarity:

The manuscript by Peterl et al. deals with the still interesting question of why influenza A viruses are filamentous in natural isolates but adopt a spherical phenotype in cell culture. The authors generated recombinant IAV reporter viruses that display identical antigenic (HA and NA) surfaces but differ in their morphology due to expression of an M1 protein that confers a spherical or filamentous phenotype. The data show that spherical viruses exhibit increased entry kinetics and spread faster in cell culture compared to filamentous viruses and that this is also the case in the presence of mucins and at a low cell density. Interestingly, the authors found that spherical viruses are more efficiently blocked by neutralizing HA antibodies than filamentous viruses, providing an interesting advantage for the filamentous phenotype of natural IAV isolates due to antibody pressure.

The manuscript is of the usual excellent quality of the working group of Petr Chlanda and the data are very interesting. The experiments are well thought out and the results are comprehensible, convincing and visually very clear. The fact that a current preprint also describes that neutralizing antibodies drives filamentous virus formation (as mentioned by the authors in the discussion) does not diminish the message and quality of this work.

There were a few minor open questions that came to mind that could be included in the discussion:

The authors found that the filamentous morphology was stable throughout multiple rounds of infection during plaque formation. Is this still the case even with multiple passages (e.g 10x) in cell culture or does the number of spherical particles increase at some point?

Thank you for your positive feedback and this suggestion. We performed serial passaging of WSN and WSN-M1_{Udom} in MDCK cells in the presence of 1 nM MEDI8852 antibody and harvested supernatants from passage 1 and 5. Supernatants were plunge-frozen, and virion counts and morphologies were determined by cryo-electron microscopy. Data from at least 80 analyzed virions per condition showed that the overall number of spherical and filamentous virions was reduced after passage 5 under antibody pressure (Fig 5 D). However, both morphologies remained stable throughout five passages in the presence of MEDI8852 (Fig. 5 E). We did not observe an increase in spherical particles after five passages.

The filamentous virus spreads slower in cell culture. Does NA play a role here? NA is probably distributed differently on the surface of filamentous viruses (at the tips) than on spherical viruses?

Thank you for this comment. As correctly pointed out, NA is enriched on one side/tip of filamentous (Calder et al., 2010, doi:10.1073/pnas.1002123107) or spherical IAV as now highlighted in Figure 1 D and E (white arrowheads). This asymmetric NA distribution and the HA-NA balance have been reported to be crucial for the release of newly formed virions and their spread through the mucus layer in the airway epithelium (De Vries et al., 2019, doi: [10.1016/j.tim.2019.08.010](https://doi.org/10.1016/j.tim.2019.08.010)). Additionally, we compared the role of NA in the spread of spherical and filamentous IAV by performing fluorescent plaque assays in the presence of

Full Revision

Zanamivir, a potent NA inhibitor. Analysis of plaque growth in the presence of increasing Zanamivir concentrations showed that the spread of both IAV morphologies was inhibited to a comparable extent (Figure 4 F and extended Figure 4 C). This result suggests that the inhibition of NA enzymatic activity does not influence the IAV morphology-dependent spread. We have included this information in the results (Line 281-285) and discussion (Line 465-468).

Reviewer #3 (Significance (Required)):

The manuscript is of the usual excellent quality of the working group of Petr Chlanda and the data are very interesting. The experiments are well thought out and the results are comprehensible, convincing and visually very clear.

Dear Petr,

Thank you for submitting your fully revised Review Commons manuscript. We have now received input from all original reviewers, who find that their main concerns have been addressed satisfactorily and recommend acceptance of the manuscript.

In addition to the minor comments by reviewer #2, there remain a few editorial points that need addressing before I can extend official acceptance of the manuscript:

1. Please submit the manuscript text file in a .docx format without track changes. The figures should be uploaded as individual production quality figure files in the .eps, .tif, or .jpg format (one file per figure). Figure legends should remain in the manuscript text file.

1. We can accommodate up to five Expanded View (EV) figures. Therefore, I would recommend uploading the figures currently included in Supplemental Material file as individual Figure files with nomenclature Figure EV1-EV5 with the appropriate callouts updated in the manuscript text and legends listed below main figure legends in the manuscript file.

2. Please make sure that the order of the sections in the manuscript is as follows: abstract, introduction, results, discussion, materials & methods, data availability section, acknowledgments, disclosure statement and competing interests, references, main figure legends, tables, expanded figure legends.

3. Please reduce the number of keywords to maximally five.

4. Please check that the funding information is correct and identical both in the manuscript and our online system. Currently, Max Planck School "Matter to Life" funded by the German Federal Ministry of Education and Research (BMBF) is missing in our online system. Funding section should be included in Acknowledgements section

5. Please submit a complete author checklist, which you can download from our author guidelines (<https://www.embopress.org/pb-assets/embo-site/EMBO%20Press%20Author%20Checklist-1642513524327.xlsx>). Please insert information in the checklist that is also reflected in the manuscript. The completed author checklist will also be part of the Review Process File.

6. CRedit has replaced the traditional author contributions section because it offers a systematic, machine-readable author contributions format that allows for more effective research assessment. Please remove the Authors Contributions from the manuscript and use the free text boxes beneath each contributing author's name in our online submission system to add specific details on the author's contribution. More information is available in our guide to authors.

7. All Materials and Methods need to be described in the main text using our 'Structured Methods' format.

According to this format, the Methods section includes a Reagents and Tools Table (listing key reagents, experimental models, software and relevant equipment and including their sources and relevant identifiers) followed by a Methods and Protocols section describing the methods, ideally using a step-by-step protocol format. The aim is to facilitate adoption of the methodologies across labs.

Please download and fill our Reagents and Tools Table template (.docx), which you can find in our author guidelines:

<https://www.embopress.org/page/journal/14602075/authorguide#structuredmethods>

8. Issues with figure callouts: all callouts should be listed sequentially. Supplementary/EV figures are missing callouts in the manuscript text - there is only a callout for Suppl. Figures 4 and 5. Figure panels Fig. 2C-D, 2G-J, 3F, 3J are not mentioned in the text, while there are callouts for the missing panels for Fig. 4I-N.

9. In the Data Availability section, please add a resolvable link to the electron tomography dataset. More information about the format of this section can be found here: <https://www.embopress.org/page/journal/14602075/authorguide#dataavailability>.

10. In our standard image integrity check, we noted that the following figure panels are reused in the manuscript:

- between figure 1G and Extended Figure 1F;
- between Figure 4G and Extended Figure 4N.

If this was intentional, please mention the reuse in the figure legend.

11. Our data editors have flagged the following issues in figure legends that need correcting:

- Please note that the legend for figure 2 is not provided in the sequential manner. This needs to be corrected.
- Please provide the exact p values in the legends of figures 1I; extended figures 1E, 3B, 5.
- Please indicate the statistical test used for data analysis in the legend of extended figure 1E.
- Please define the box plots in terms of centre, bounds of box and whiskers, and percentile in the legends of extended figure 3B
- Please define the box plots in terms of minima, maxima, centre, bounds of box and whiskers, and percentile in the legends of extended figure 3B, 4L, M.
- Please provide information on the number and nature of replicates in the legends of figures 4B, 5A; extended figures 4K-M.
- Please define the error bars in the legends of figures 3A, C, D; Extended figure 1E.
- Please define the scale bar for figures 3E-H.

12. Papers published in The EMBO Journal are accompanied online by a 'Synopsis' to enhance discoverability of the manuscript. It consists of A) a short (1-2 sentences) summary of the findings and their significance, B) 3-4 bullet points highlighting key results and C) a synopsis image that is 550x300-600 pixels large (width x height, jpeg or png format). You can

either show a model or key data in the synopsis image. Please note that the image size is rather small and that text needs to be readable at the final size.

With best wishes,

Ieva

Ieva Gailite, PhD
Senior Scientific Editor
The EMBO Journal
Meyerohofstrasse 1
D-69117 Heidelberg
Tel: +4962218891309
i.gailite@embojournal.org

Revision to The EMBO Journal should be submitted online within 90 days, unless an extension has been requested and approved by the editor; please click on the link below to submit the revision online before 10th Jul 2025:

Link Not Available

Referee #1:

The manuscript has been well revised and the authors have addressed all my criticisms and questions. I have no further comments.

Referee #2:

The revised work explores a long standing question in the influenza field (the factors governing filamentous particle morphology). The manuscript presents a set of novel and convincing claims that are appropriately discussed in the context of earlier literature. The study is of broad interest to virologists and cell biologists, and employs novel and rigorous techniques (electron microscopy and imaging of viral spread in living cells) to look at the morphology of authentically filamentous viruses in a genetically elegant manner. The work added to the revised manuscript has satisfied my concerns. I identified a few minor typos (listed below) that should be checked and corrected before publication.

- There appears to be a mistake in labeling with Extended Figure 1H/I- the authors refer to IF staining for NP or M2, but both of these panels contain M2 labels (see also F and G).
- Figure 5D does not appear to have any data verifying expression of MUC5AC in Calu3 cells, but is instead a graph of virus counts after passaging with neutralizing antibody. After a bit of digging, I see the data in Figure 4D though, so that looks fine.

Referee #3:

This revised manuscript by Peterl and colleagues seeks to understand the long-standing observation that influenza A virus generally exhibits a filamentous phenotype *in vivo* which is lost upon serial passaging *in vitro* or in embryonated chicken eggs. In addressing this question, the authors perform a detailed quantitative comparison of how filamentous and spherical strains of influenza spread in cell culture in the presence or absence of perturbations including neutralizing antibodies, mucin, and disruption of cell-cell junctions.

The manuscript reports several observations that will be of interest to researchers in the area of influenza virus morphology and

spread. Using a combination of imaging modalities, the authors convincingly demonstrate that spherical strains of influenza virus produce larger plaques than filamentous strains that are isogenic except for mutations in M1. The authors show that this is at least partly attributable to differences in entry kinetics. The authors also recapitulate a prior finding that filamentous viruses are more resistant to neutralizing antibodies than spherical ones.

The revisions to this manuscript have addressed my concerns regarding the clarity of experimental approaches and the discussion of technical limitations of the study.

Rev_Com_number: RC-2024-02635

New_manu_number: EMBOJ-2025-120718-T

Corr_author: Chlanda

Title: Morphology-dependent entry kinetics and spread of influenza A virus

Dear Reviewers,

We would like to thank you for the feedback and are pleased to submit the revised version of our manuscript.

Referee #1:

The manuscript has been well revised and the authors have addressed all my criticisms and questions. I have no further comments.

Thank you very much for your time. Your suggestions have been very valuable in improving the quality of our manuscript.

Referee #2:

The revised work explores a long standing question in the influenza field (the factors governing filamentous particle morphology). The manuscript presents a set of novel and convincing claims that are appropriately discussed in the context of earlier literature. The study is of broad interest to virologists and cell biologists, and employs novel and rigorous techniques (electron microscopy and imaging of viral spread in living cells) to look at the morphology of authentically filamentous viruses in a genetically elegant manner. The work added to the revised manuscript has satisfied my concerns. I identified a few minor typos (listed below) that should be checked and corrected before publication.

- There appears to be a mistake in labeling with Extended Figure 1H/I- the authors refer to IF staining for NP or M2, but both of these panels contain M2 labels (see also F and G).
- Figure 5D does not appear to have any data verifying expression of MUC5AC in Calu3 cells, but is instead a graph of virus counts after passaging with neutralizing antibody. After a bit of digging, I see the data in Figure 4D though, so that looks fine.

We appreciate Reviewer 2's valuable comments. Regarding the two minor points:

- We apologize for the confusion caused by the labeling error in Extended Figure 1H/I thank you for bringing it to our attention. We have corrected this labeling mistake for all panels and the figure legend to accurately reflect M2.
- We apologize for the incorrect reference to Figure 5D. As you pointed out, the data verifying MUC5AC expression in Calu3 cells is actually presented in Figure 4D, not 5D. We have corrected the manuscript to accurately reflect this and thank you for pointing out the inconsistency.

Referee #3:

This revised manuscript by Peterl and colleagues seeks to understand the long-standing observation that influenza A virus generally exhibits a filamentous phenotype in vivo which is lost upon serial passaging in vitro or in embryonated chicken eggs. In addressing this question, the authors perform a detailed quantitative comparison of how filamentous and spherical strains of influenza spread in cell culture in the presence or absence of perturbations including neutralizing antibodies, mucin, and disruption of cell-cell junctions.

The manuscript reports several observations that will be of interest to researchers in the area of influenza virus morphology and spread. Using a combination of imaging modalities, the authors convincingly demonstrate that spherical strains of influenza virus produce larger

plaques than filamentous strains that are isogenic except for mutations in M1. The authors show that this is at least partly attributable to differences in entry kinetics. The authors also recapitulate a prior finding that filamentous viruses are more resistant to neutralizing antibodies than spherical ones.

The revisions to this manuscript have addressed my concerns regarding the clarity of experimental approaches and the discussion of technical limitations of the study.

We are grateful for your careful consideration of our manuscript, and we appreciate your positive remarks and the absence of further comments. Your previous suggestions have certainly enhanced the clarity and quality of our work, and we truly appreciate your contribution to the review process.

We hope that these revisions address your concerns, and we greatly appreciate your detailed review.

Best wishes,

Petr Chlanda

on behalf of all co-authors

Heidelberg, April 25th, 2025

Dear Petr,

Thank you for addressing the final editorial issues. I sincerely apologise for the delay in the processing of your manuscript due to conference attendance last week and the resulting backlog. I am now pleased to inform you that your manuscript has been accepted for publication.

I will now look into the synopsis text that you kindly provided and will let you know if any textual edits to the journal style are needed before we forward your manuscript to our publishers.

If you have any questions, please do not hesitate to contact the Editorial Office. Thank you for this contribution to The EMBO Journal and congratulations on a nice study!

Best wishes,

Ieva

Ieva Gailite, PhD
Senior Scientific Editor
The EMBO Journal
Meyerohofstrasse 1
D-69117 Heidelberg
Tel: +4962218891309
i.gailite@embojournal.org

Rev_Com_number: RC-2024-02635
New_manu_number: EMBOJ-2025-120718R
Corr_author: Chlanda
Title: Morphology-dependent entry kinetics and spread of influenza A virus